# The Lamin B receptor is essential for cholesterol synthesis and perturbed by disease-causing mutations

**Pei-Ling Tsai[1†], Chenguang Zhao[1†], Elizabeth Turner[1], Christian Schlieker[1,2]\***

[1]Department of Molecular Biophysics and Biochemistry, Yale University, New Haven, United States; [2]Department of Cell Biology, Yale School of Medicine, New Haven, United States

**Abstract** Lamin B receptor (LBR) is a polytopic membrane protein residing in the inner nuclear membrane in association with the nuclear lamina. We demonstrate that human LBR is essential for cholesterol synthesis. LBR mutant derivatives implicated in Greenberg skeletal dysplasia or Pelger-Huët anomaly fail to rescue the cholesterol auxotrophy of a LBR-deficient human cell line, consistent with a loss-of-function mechanism for these congenital disorders. These disease-causing variants fall into two classes: point mutations in the sterol reductase domain perturb enzymatic activity by reducing the affinity for the essential cofactor NADPH, while LBR truncations render the mutant protein metabolically unstable, leading to its rapid degradation at the inner nuclear membrane. Thus, metabolically unstable LBR variants may serve as long-sought-after model substrates enabling previously impossible investigations of poorly understood protein turnover mechanisms at the inner nuclear membrane of higher eukaryotes.

**\*For correspondence:** christian. schlieker@yale.edu

[†]These authors contributed equally to this work

**Competing interests:** The authors declare that no competing interests exist.

## Introduction

Lamin B receptor (LBR) is an evolutionary conserved, multifunctional protein (*Olins et al., 2010*). The N-terminal moiety of LBR, which resides in the nucleoplasm, contains a chromatin-binding TUDOR domain and associates with the nuclear lamina (*Makatsori et al., 2004*; *Pyrpasopoulou et al., 1996*; *Worman et al., 1988*), while the polytopic C-terminal domain anchors LBR in the inner nuclear membrane (INM) and has sequence homology to sterol C14 reductases (*Li et al., 2015*; *Silve et al., 1998*; *Worman et al., 1990*). Sterol C14 reductases are widely conserved throughout evolution and are responsible for the reduction of a specific carbon-carbon double bond as part of the tightly controlled enzymatic cascade which results in the production of cholesterol and cholesterol-related compounds (*Benveniste, 2004*; *Holmer et al., 1998*) (*Figure 1*).

There is currently no known functional connection between the chromatin-binding N-terminus of LBR and the sterol reductase C-terminus of the protein. The function of the sterol reductase domain (SRD) of LBR is further obfuscated by the fact that human cells have a second C14 sterol reductase enzyme called TM7SF2, which is conserved in evolution and localizes to the endoplasmic reticulum (ER), where other enzymes responsible for cholesterol biogenesis are typically found (*Bennati et al., 2006*). The TM7SF2 promoter harbors a sterol response element (SRE) common to most, if not all enzymes implicated in cholesterol synthesis, allowing for the tight regulation of their transcription in response to cholesterol availability (*Brown and Goldstein, 1999*; *Sharpe and Brown, 2013*). However, the LBR gene lacks an SRE consensus sequence and is constitutively expressed (*Bennati et al., 2006*; *Cohen et al., 2008*; *Sharpe and Brown, 2013*), drawing into question whether LBR has a significant role in cholesterol synthesis.

**eLife digest** In humans, mutations in the gene that encodes a protein called Lamin B receptor can lead to diseases ranging from harmless anomalies of blood cells to fatal developmental defects. The severity of the disease depends on the nature of the specific mutation, and whether one or both copies of the gene are affected. Lamin B receptor – or LBR for short – is found at the envelope that surrounds the cell's nucleus and was previously proposed to anchor this envelope to an underlying scaffold to provide it with support. LBR can also catalyze a chemical reaction involved in producing cholesterol – an essential component of cell membranes. However, this enzymatic activity was assumed to be less important because a second enzyme named TM7SF2 can perform the same reaction. Thus, it was not clear – at the molecular level – why the mutations in this gene lead to a variety of diseases.

All disease-causing mutations map to the part of LBR that is responsible for its enzymatic activity. This fact motivated Tsai, Zhao et al. to reassess the importance of LBR for the production of cholesterol. The experiments revealed that many human cells that can be grown in the laboratory strictly depend on LBR to produce cholesterol. As such, these findings challenge the previous assumption that TM7SF2 can compensate for the loss of LBR's activity and sustain cholesterol synthesis.

Tsai, Zhao et al. also discovered that all known disease-causing mutations strongly perturb LBR's ability to engage in cholesterol synthesis, albeit through different mechanisms. Some mutations interfered with the enzyme ability to bind with an essential molecule or cofactor that is required to catalysis; others led to LBR rapidly degrading at the nuclear envelope.

It was previously not known that proteins could be degraded at the inner membrane of the nuclear envelope of mammalian cells, and LBR mutants may turn out to be useful tools to investigate how this happens in future. Further studies could also test if other diseases caused by mutations in proteins found in the nuclear envelope act in similar ways, or if mutations in these proteins inhibit the nucleus's protein disposal machinery.

While congenital diseases associated with defects in cholesterol homeostasis have been extensively investigated (*Goldstein and Brown, 2015*), much less is known about the possible involvement of *LBR* mutations in cholesterol metabolism. Two congenital disorders are known to be associated with mutations in LBR: Pelger-Huët anomaly and Greenberg skeletal dysplasia (*Oosterwijk et al., 2003*; *Shultz et al., 2003*; *Wassif et al., 2007*; *Waterham et al., 2003*) (see *Table 1*). Pelger-Huët anomaly is an autosomal dominant disorder in which a single mutation in one LBR allele results in abnormal hypolobulation of granulocyte nuclei (*Best et al., 2003*; *Hoffmann et al., 2002*; *Shultz et al., 2003*). The other human disease associated with LBR, Greenberg skeletal dysplasia, is a perinatally lethal, autosomal recessive condition that results in abnormal bone development, fetal hydrops, and the ultimate nonviability of the fetus (*Chitayat et al., 1993*; *Greenberg et al., 1988*; *Horn et al., 2000*; *Konstantinidou et al., 2008*; *Trajkovski et al., 2002*). Interestingly, mounting evidence indicates that Greenberg skeletal dysplasia results from the inheritance of two mutant *LBR* alleles that when heterozygous cause Pelger-Huët anomaly (*Konstantinidou et al., 2008*; *Oosterwijk et al., 2003*), indicating that the two diseases represent different allelic states of the same chromosomal lesion. However, it is unclear whether these diseases are caused by structural changes in the nuclear lamina, or whether they are diseases of cholesterol metabolism (*Clayton et al., 2010*; *Olins et al., 2010*; *Wassif et al., 2007*; *Waterham et al., 2003*; *Worman and Bonne, 2007*).

In this study, we show that LBR is essential for cholesterol synthesis. Using a human cell culture model, we demonstrate that it is this function that is perturbed by LBR mutations associated with Pelger-Huët anomaly and Greenberg skeletal dysplasia, suggesting a loss-of-function mechanism for these congenital disorders. Unexpectedly, disease-causing mutations involving C-terminal truncations of LBR lead to their rapid degradation in the nuclear envelope (NE). Such LBR mutants appear to be dislocated from the INM directly into the nucleoplasm, unlike traditional substrates of the ER-associated degradation (ERAD) machinery, which are eliminated in the cytosol after their dislocation

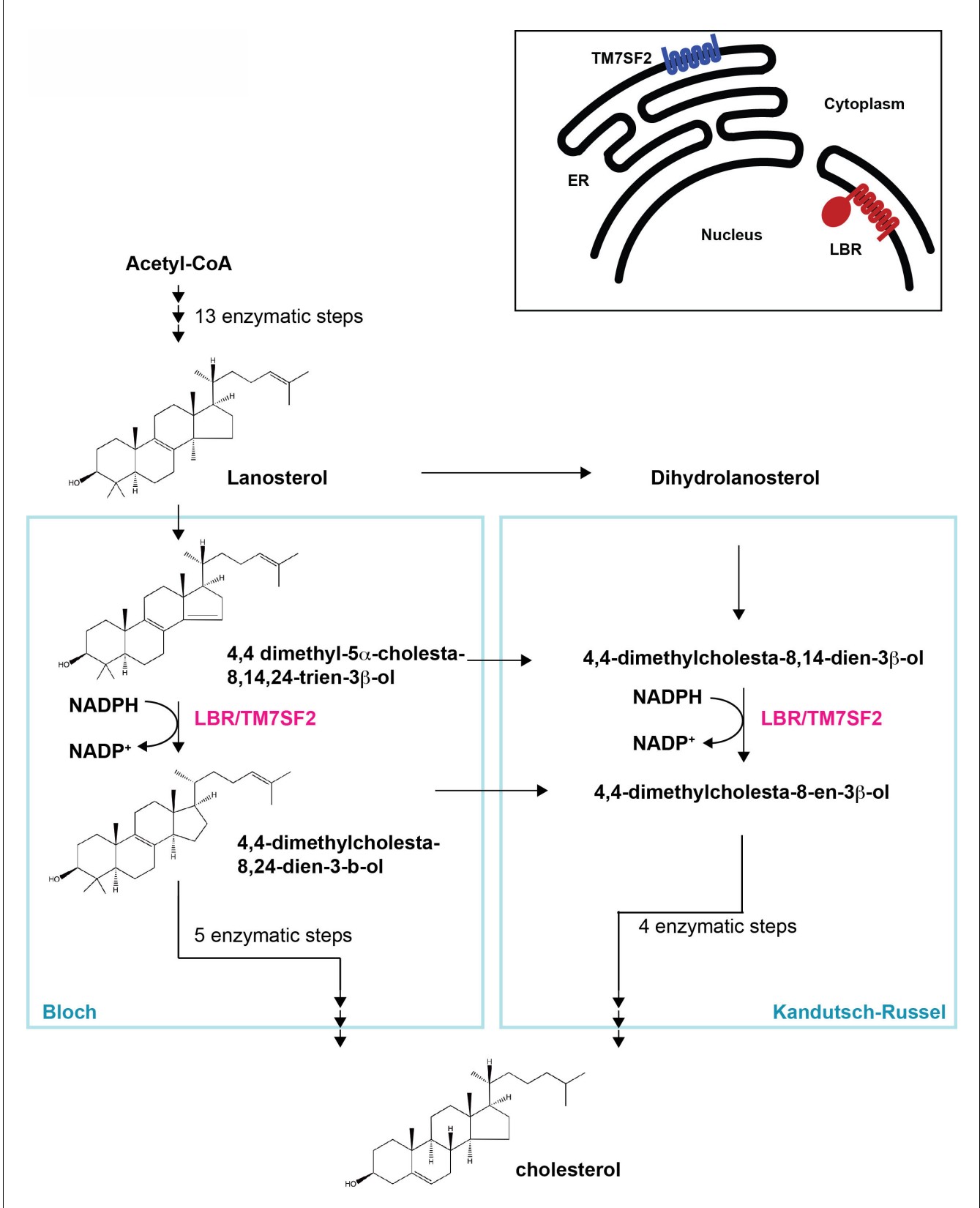

**Figure 1.** Cholesterol biosynthesis pathway. Simplified schematic of cholesterol biosynthesis starting from Acetyl-CoA. After 13 enzymatic steps, the intermediate Lanosterol can enter one of two parallel pathways designated Bloch and Kandutsch-Russel pathways, respectively, both of which employ

*Figure 1 continued*

an NADPH-dependent reduction step which can be catalyzed by sterol C14 reductases LBR or TM7SF2 (highlighted in magenta). Adapted, with modifications, from (*Sharpe and Brown, 2013*). The inset on the upper right depicts distinct subcellular localizations of the human sterol reductases LBR and TM7SF2 localizing to the inner nuclear membrane and ER, respectively.

from the ER (*Claessen et al., 2012*; *Vembar and Brodsky, 2008*). Metabolically unstable LBR mutant proteins will therefore be informative for future studies aimed at elucidating mechanisms of protein quality control at the nuclear envelope of mammalian cells, a site that was previously experimentally inaccessible due to the absence of suitable model substrates.

## Results

### Generation of LBR knockout HeLa cell lines

In order to clarify the cellular function of LBR both in cholesterol metabolism and as a structural component of the nuclear lamina, we used the CRISPR/Cas9 system (*Mali et al., 2013*) to generate LBR knockout HeLa cell lines (*Figure 2—figure supplement 1A*). CRISPR/Cas9 editing of *LBR* alleles was performed in a recombination-competent HeLa FlpIn cell line (hereafter designated wild type or WT cells), allowing for rapid and efficient introduction of WT rescue and disease-specific *LBR* alleles into the LBR knockout cell background via site-specific recombination (*Turner et al., 2015*). CRISPR/Cas9-treated WT cells were screened for the absence of full-length LBR protein by immunoblotting using antibodies against both the N and C termini of the protein (*Figure 2—figure supplement 1B*), and via genotyping using PCR primers flanking the CRISPR target site (*Figure 2—figure supplement 1A*, arrows). A clone was obtained that yielded no detectable LBR protein as judged by immunoblotting, corresponding to the absence of a PCR product of the size predicted by the wild-type *LBR* allele (*Figure 2—figure supplement 1C*), indicating that all LBR alleles had been effectively targeted.

To exclude the presence of hypomorphic alleles, we performed deep sequencing on the genetic locus encompassing the LBR CRISPR/Cas9 target site. Since HeLa cells are aneuploid, including three complete copies of chromosome 1 where the LBR gene is located, any LBR knockout should have three distinct genome 'edits'.

Indeed, sequence analysis revealed three distinct mutant alleles, all containing frame-shift mutations or premature stop codons within the 5' region of the LBR open reading frame, showing that no more than 12 amino acids of LBR WT sequence can be produced from any of the three mutant alleles (*Figure 2—figure supplement 2*).

### Deletion of LBR does not alter NE integrity

As indicated by its name, LBR has long been implicated in NE integrity and NE anchoring to the nuclear lamina (*Appelbaum et al., 1990*; *Worman et al., 1990*, *1988*; *Ye and Worman, 1994*),

**Table 1.** Diseases-associated LBR mutations used in this study.

| LBR variant | Mutation | Phenotype | Reference |
|---|---|---|---|
| N547D | c.1639A>G | Heterozygous - No Phenotype | *Clayton et al., 2010* |
| | p.N547D | Homozygous - Greenberg Dysplasia | *Konstantinidou et al., 2008* |
| R583Q | c.1748G>A | Heterozygous - No Phenotype | *Clayton et al., 2010* |
| | p.R583Q | Homozygous - Greenberg Dysplasia | |
| 1402TΔ | c.1402delT | Heterozygous - Phenotype Unknown | *Clayton et al., 2010* |
| | p.Y468TfsX475 | Homozygous - Greenberg Dysplasia | |
| 1600* | c.1599-1605TCTTCTA→CTAGAAG | Heterozygous - Pelger-Huët Anomaly | *Waterham et al., 2003* |
| | p.X534 | Homozygous - Greenberg Dysplasia | |

prompting us to investigate if removing LBR perturbs the structure and composition of the nuclear lamina. We performed immunofluorescence microscopy analysis of known INM proteins and components of the nuclear lamina in both LBR knockout (KO) and WT cells. No differences in overall cell morphology or growth were observed between WT and LBR KO cells under normal growth conditions (*Figure 2A*). Surprisingly, we found no change in the localization of Lamin B1, Lamin A/C or Emerin in LBR KO cells compared to control cells (*Figure 2A*). Similarly, we found that the absence of LBR also had no effect on the localization of other structural proteins of the NE such as Sun1 or Sun2, which serve as the INM components of the LINC (l̲inker of n̲ucleoskeleton and c̲ytoskeleton) complex (*Crisp et al., 2006*) (*Figure 2—figure supplement 3A and B*). Similar results were obtained for other NE, nuclear and ER markers (*Figure 2—figure supplement 3*).

Lastly, we utilized electron microscopy on both LBR KO and WT cells to obtain high-resolution images of the NE. We found that in both WT and LBR KO cells, the inner and outer nuclear membranes were regularly spaced and featured a nuclear lamina and nuclear pores of normal morphology (*Figure 2B*, arrowheads). These data, together with our previous observations by immunofluorescence microscopy, indicate that in these cells, LBR does not play a significant role in maintaining NE integrity. We cannot, however, exclude that LBR plays a role in the structural integrity of the nuclear lamina in specific cell types, under conditions of mechanical stress, or developmental stages found only in the context of the living organism.

## LBR is required for viability under cholesterol starvation conditions

Next, we set out to investigate the role of LBR in cholesterol synthesis. In order to determine if LBR is required for cell proliferation under cholesterol-restrictive growth conditions, HeLa WT cells and LBR KO cells were cultured in medium containing lipoprotein-depleted fetal bovine serum. We found that after 4 days under cholesterol-restrictive growth conditions LBR KO, but not WT HeLa cells exhibited slow growth, cell rounding, and detachment, followed quickly by cell death on days 5–7 (*Figure 2C*). Addition of 10 µM exogenous cholesterol to the cell culture medium effectively rescued the observed sensitivity of LBR KO cells to low-cholesterol growth conditions (*Figure 2C*), indicating that the observed phenotype is in fact due to a deficiency in cholesterol production. The observed growth defect of LBR KO cells was also rescued by the addition of low density lipoprotein (LDL) particles, a physiologically relevant cholesterol carrier (*Lodish, 2013*), to the cell culture medium (*Figure 2C*). These results suggest that the primary function of LBR in our tissue culture model is to sustain cholesterol biogenesis when the extracellular supply of cholesterol is scarce.

Given that LBR is endowed with sterol C14 reductase activity (*Bennati et al., 2006*; *Silve et al., 1998*), we tested directly whether LBR-deficient cells are compromised in *de novo* cholesterol synthesis. To this end, HeLa WT and LBR KO cells were cultured in lipid-deprived medium for 48 hr, followed by the addition of $^{14}$C acetate to the culture medium. After four hours, cells were harvested, lysed and the extracted lipids were separated via thin layer chromatography (TLC). We readily detected newly synthesized cholesterol in WT cells, as judged by its co-migration with purified $^{14}$C cholesterol, which was included as a standard (*Figure 2D*). Notably, we observed a near-complete loss of cholesterol synthesis in LBR KO cells (*Figure 2D*), validating our previous assumption that LBR KO cells cannot effectively sustain cholesterol synthesis.

Finally, we asked whether these findings can be reproduced in other human cell types. We chose human foreskin fibroblasts (HFFs) to include non-transformed primary cells, as well as commonly used HEK293T cells. In either case, we observed a strong reduction of viability in LBR-silenced cells under cholesterol-restrictive conditions, whereas control cells transfected with non-targeting siRNAs displayed normal cell morphology under these conditions (*Figure 3 A,B,D,E*). To validate our RNA interference approach, we subjected the corresponding cell lysates to immunoblotting and observed a robust LBR knockdown efficacy in both cell types (*Figure 3C,F*). These findings rule out that the cholesterol auxotrophy observed in HeLa LBR KO cells is attributable to their degeneracy or their transformed nature.

## LBR is required for cholesterol synthesis despite the presence of TM7SF2

Our assignment of LBR to an essential role in cholesterol synthesis contrast earlier findings in mice, which reported redundant functions for LBR and TM7SF2 in cholesterol synthesis (*Wassif et al.,*

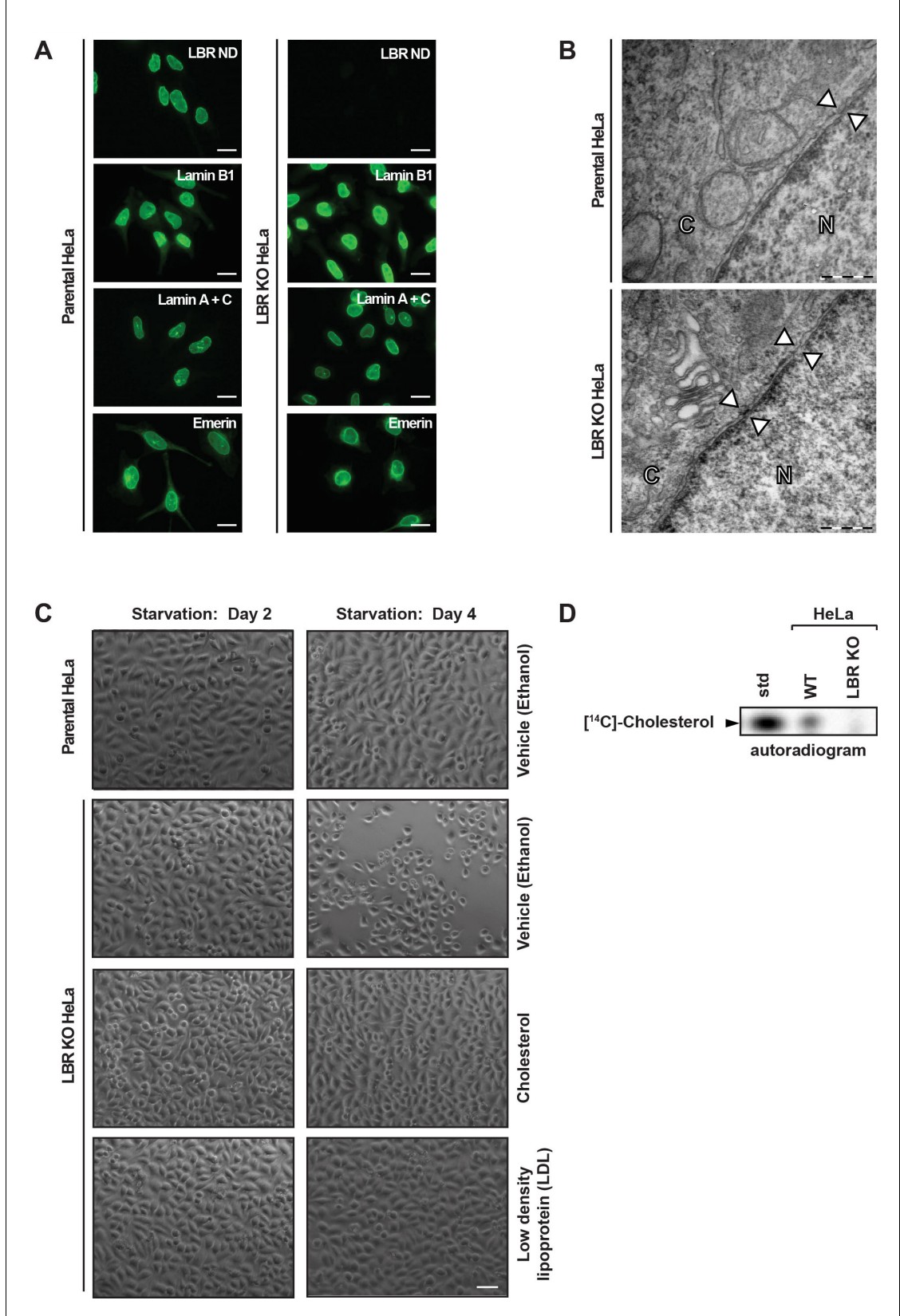

**Figure 2.** LBR deficient cells display normal nuclear envelope morphology but are sensitive to cholesterol restriction. (**A**) HeLa WT and HeLa LBR KO cells were stained with antibodies against LBR using an antibody recognizing the N-terminal domain (ND), Lamin B1, Lamin A/C, and Emerin and then

*Figure 2 continued on next page*

*Figure 2 continued*

imaged by immunofluorescence microscopy. Scale bar = 10 µm. (B) EM images of WT and LBR KO HeLa cells. The nucleus (N) and cytosol (C) are labeled and nuclear pores are indicated by arrowheads. Scale bar = 500 nm. (C) Indicated cell lines were cultured under cholesterol-restrictive growth conditions for two days, imaged by bright-field microscopy, and cultured for 2 more days under cholesterol restrictive conditions, and imaged again. Cells were exposed either to vehicle, free cholesterol or LDL (10 µM) for four days as indicated. Scale bar = 50 µm. (D) HeLa WT and LBR KO cells in LPDS-containing medium were metabolically labeled with [$^{14}$C]-acetate. Lipids were extracted and separated by TLC and visualized via autoradiography. [$^{14}$C]-cholesterol was included as a standard (std).

The following figure supplements are available for figure 2:

**Figure supplement 1.** LBR knockout cells were generated using the CRISPR/Cas9 genome editing system.

**Figure supplement 2.** The genomic LBR CRISPR target site of LBR KO HeLa cells was analyzed using Illumina MiSeq.

**Figure supplement 3.** LBR deficient cells display normal nuclear envelope morphology.

*2007*). We therefore asked whether HeLa, HEK293T and HFF cells expressed both TM7SF2 and LBR under normal and cholesterol-restrictive conditions. For this analysis, we isolated total RNA and prepared protein extracts from these three cell types grown under normal and cholesterol-restrictive conditions. TM7SF2 and LBR transcript levels were quantified via qPCR, and the observed fold change under restrictive conditions was normalized relative to the transcript abundance under normal growth conditions, which was set to one. As expected (*Bennati et al., 2006*), we found that HFF and HeLa cells upregulated TM7SF2 under starvation conditions (*Figure 3G*), whereas LBR was constitutively expressed and unresponsive to cholesterol starvation (*Figure 3H*). Interestingly, HEK293T cells did not materially up-regulate TM7SF2 on the transcript level (*Figure 3G*). Similar results were obtained when we monitored the LBR and TM7SF2 protein levels via immunoblotting. The abundance of the TM7SF2 protein in HFF and HeLa cells increased under cholesterol-restrictive conditions, whereas HEK293T cells displayed somewhat higher levels of TM7SF2 even under normal conditions but did not upregulate TM7SF2 to the same degree as HFF or HeLa cells (*Figure 3I*).

Lastly, we wanted to ascertain that LBR KO cells are capable of mounting a sterol regulatory element-binding protein-2 (SREBP2)-dependent transcriptional response under cholesterol starvation conditions (*Brown and Goldstein, 1997*). We did not observe significant differences between HeLa WT and HeLa LBR KO cells in their ability to up-regulate TM7SF2 on the transcript level, as judged by qPCR (*Figure 3—figure supplement 1A*). Similar results were obtained for HMG-CoA reductase (*Figure 3—figure supplement 1C*), which was included as a control since the HMG-CoA reductase gene is an established target of SREBP2 (*Brown and Goldstein, 1997*). In accordance with previous findings (*Bennati et al., 2006*), the observed upregulation was dependent on SREBP2, as judged by a profound reduction of TM7SF2 and HMG-CoA reductase on the transcript levels in cells depleted of SREBP2 via RNA interference.

Finally, to exclude the formal possibility that coding mutations in TM7SF2 are responsible for the observed essential role of LBR in HeLa cells through perturbation of TM7SF2 sterol C14 reductase activity, we cloned and sequenced the corresponding cDNA. This isolated cDNA (1401 bp) contains the entire coding sequence (CDS) of the published transcript variant 1 (NM_003273), with no apparent mutations as evidenced by a sequence alignment to the CDS of NM_003273 (*Figure 3—source data 1*). This isoform appears to be the major isoform expressed in HeLa, HFF, and HEK 293T cells, as judged by the observed co-migration of the corresponding TM7SF2 proteins in SDS-PAGE/Immunoblots (*Figure 3I*).

Based on the foregoing, we arrive at the conclusion that LBR is essential for cholesterol synthesis in several human cell lines despite the presence of TM7SF2.

## LBR disease alleles do not rescue the cholesterol auxotrophy of LBR knockout cells

Having demonstrated that LBR is essential for cell viability under cholesterol-depleted conditions, we asked whether the mutant variants of LBR found in Pelger-Huët anomaly and Greenberg skeletal dysplasia can sustain cholesterol biogenesis in human cells. To this end, we used the FlpIn gene

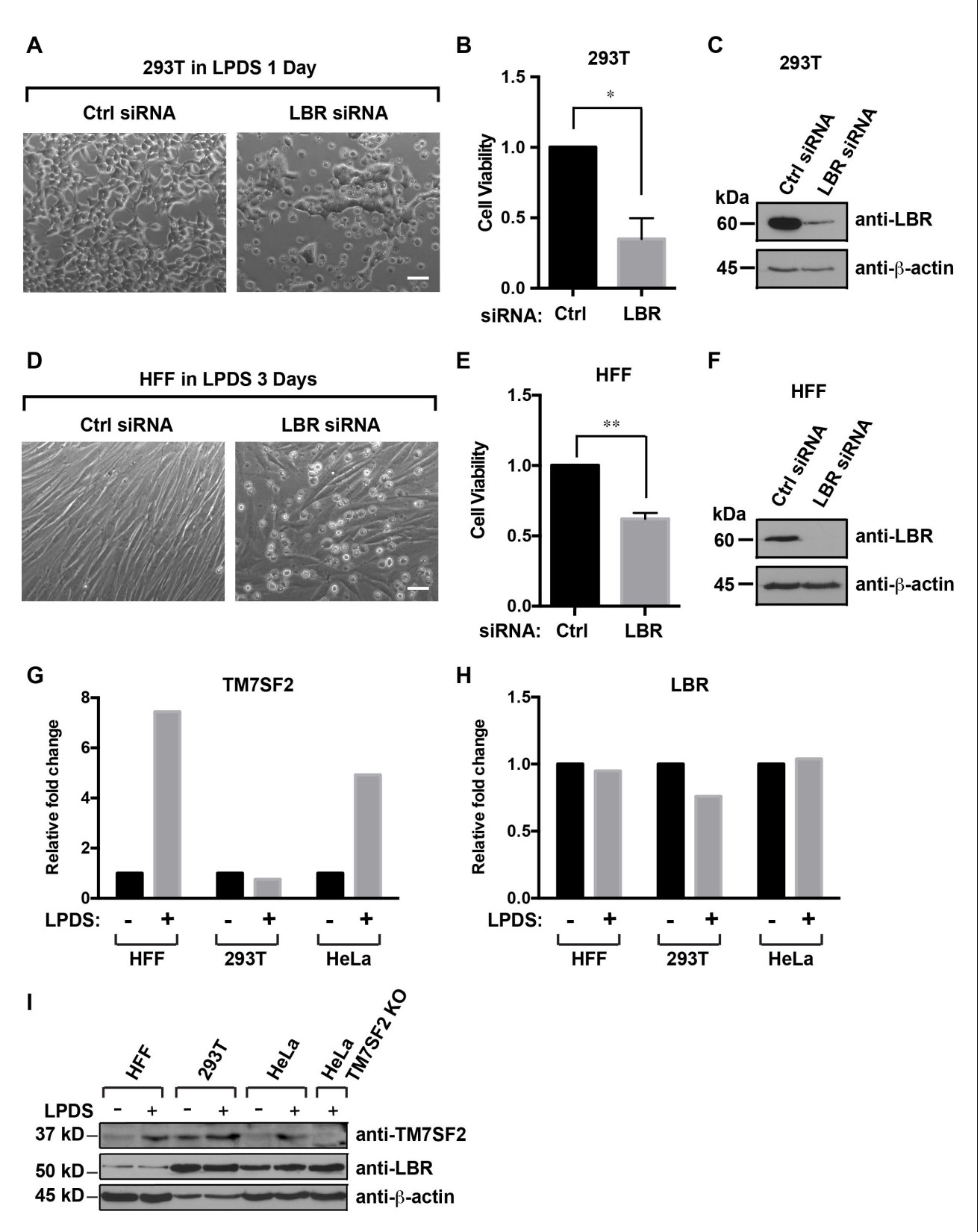

**Figure 3.** Cells with reduced levels of LBR are sensitive to cholesterol restriction despite the presence of TM7SF2 . (**A**) 293T cells were transfected with control siRNA or LBR siRNA, incubated for 48 hr, cultured in cholesterol-restrictive condition for 1 day and imaged via bright-field microscopy. Scale bar

*Figure 3 continued on next page*

*Figure 3 continued*

= 50 µm (B) Surviving adherent cells were quantified using crystal violet staining, and data were standardized relative to the level of control siRNA. The error bars represent mean ± SD from three independent experiments (N = 3), and the statistical analysis was performed using paired t-test (*p value < 0.05, **p value < 0.01). (C) An additional parallel well of 293T cells with same treatment as described above was lysed and analyzed using immunoblotting. (D–F) HFF cells were treated with siRNA and cultured under cholesterol-restrictive conditions as described in (A) with an exception that HFF cells are cultured in the LPDS medium for 3 days. (G and H) HFF, 293T, and HeLa cells were cultured in normal medium or cholesterol-restrictive medium for 2 days, and cells were harvested and split into two aliquots. Total RNAs were extracted from one aliquot, reverse transcribed into cDNA, and analyzed using real-time PCR with gene specific primers as indicated. The data was represented as a relative level to the normal condition (i.e. –LPDS was set to one in each cell line). (I) The other aliquot of cells was lysed with sample buffer and subjected to immunoblotting analysis.

The following source data and figure supplement are available for figure 3:

**Source data 1.** TM7SF2 sequence cloned using RT-PCR from HeLa cells.

**Figure supplement 1.** LBR transcription does not change under cholesterol-restrictive condition.

integration system to generate HeLa cells that express either WT LBR or LBR disease alleles under doxycycline control from the FlpIn locus (*Turner et al., 2015*). We focused on two LBR point mutations, one frameshift mutation, and one nonsense mutation, the latter of which results in a truncated LBR C-terminus (*Table 1*) (*Clayton et al., 2010*; *Konstantinidou et al., 2008*; *Waterham et al., 2003*). The two point mutations, LBR N547D and LBR R583Q (*Figure 4A*), are both associated with Greenberg skeletal dysplasia when both LBR alleles are mutated (*Clayton et al., 2010*; *Konstantinidou et al., 2008*). Both positions are highly conserved in homologous C14 sterol reductases (*Figure 5—figure supplement 1*) and are not associated with Pelger-Huët anomaly in heterozygous individuals (*Table 1*) (*Clayton et al., 2010*). For the LBR truncations, the first frameshift mutation results from a single nucleotide deletion in the C-terminal sterol reductase domain of LBR, c.1402delT, which results in the alteration of downstream codons 468–475 and the formation of a premature stop codon at position 475 (p.Y468TfsX475) (*Clayton et al., 2010*), which we designated LBR 1402TΔ. This mutation results in a C-terminally truncated LBR protein that is missing the final three membrane spanning helices that form the sterol-reductase domain of LBR (*Figure 4A*) and results in Greenberg skeletal dysplasia in homozygous individuals. It is unknown whether individuals heterozygous for the LBR c.1402delT mutation exhibit Pelger-Huët anomaly. The second frameshift mutation causing a distinct C-terminal truncation, LBR c.1599-1605TCTTCTA->CTAGAAG (LBR p. X534), which we have designated as LBR 1600*, has been shown to cause Pelger-Huët anomaly in heterozygous individuals as well as Greenberg skeletal dysplasia when both LBR alleles are mutated (*Waterham et al., 2003*). This mutation is a seven nucleotide substitution beginning at LBR position 1599 that directly causes the introduction of a premature stop codon at position LBR p.534, resulting in a truncated LBR C-terminus lacking the final two transmembrane helices of the protein (*Figure 4A*).

To determine if these mutations are loss-of-function alleles, we expressed each of these as well as wild-type LBR under doxycycline control in our LBR KO HeLa cell line. Cells expressing various LBR alleles in an LBR knockout background were grown under cholesterol-restrictive culture conditions for 7 days. We then imaged and counted the cells and measured total cholesterol content (see Materials and methods). We found that expression of LBR WT from the FlpIn locus completely rescued the growth defect observed for LBR KO cells in cholesterol starvation growth medium (*Figure 4B and C*). This excluded off-target effects as a cause for the observed phenotype. Importantly, all four disease-associated LBR alleles failed to rescue the observed growth defect (*Figure 4B and C*). LBR KO cells were found to have approximately 40% less total cholesterol content than LBR WT cells after 7 days of cholesterol starvation (*Figure 4E*). Cellular cholesterol content was fully restored to that of WT cells by expression of LBR WT from the FlpIn locus even at somewhat lower LBR expression levels compared to WT cells (*Figure 4D and E*), but not by expression of any of the four disease-associated LBR alleles (*Figure 4E*). From these data we conclude that LBR KO HeLa cells are strongly compromised for cholesterol production. Neither cell growth nor cellular cholesterol content are restored by the expression of LBR N547D, LBR R583Q, LBR 1402TΔ or LBR 1600*

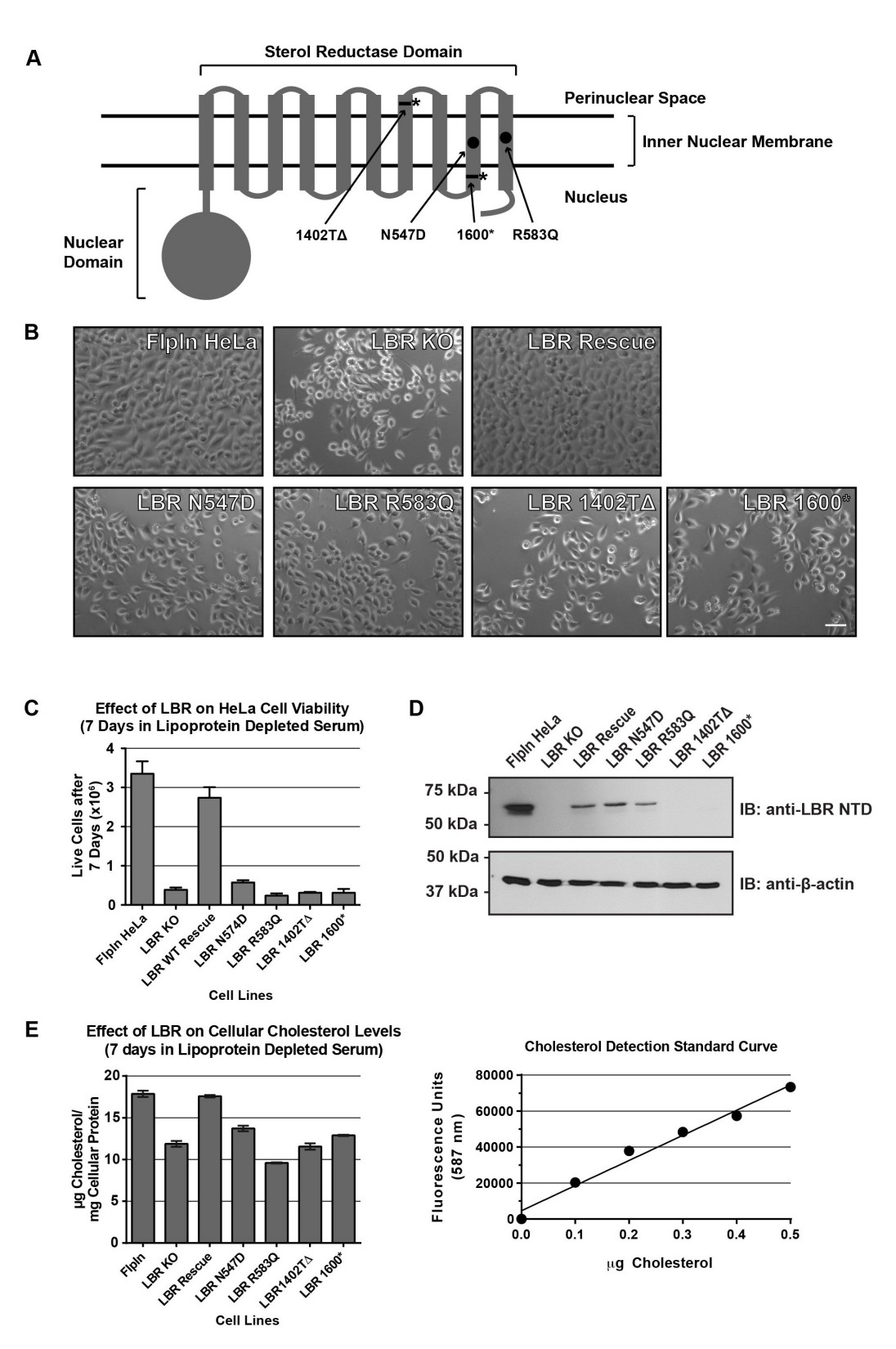

**Figure 4.** Cholesterol auxotrophy of LBR KO cells is rescued by wild-type but not disease-mutant LBR . (**A**) Domain structure of LBR. Locations of disease-associated point mutations are indicated as black circles and asterisks demark disease-associated frameshift/truncation mutations. (**B**) Parental

*Figure 4 continued on next page*

*Figure 4 continued*

WT, LBR KO, or LBR KO cells expressing either WT LBR or mutant LBR from the FlpIn locus were cultured for 7 days in cholesterol-restrictive growth medium and then imaged by bright field microscopy. (C) The cell lines described above were grown in triplicate for 7 days in cholesterol-restrictive growth conditions, trypsinized, and counted. Values represent a mean of three independent experiments with error bars indicating the standard deviation (D) Immunoblot analysis of cell lysates harvested on day 7 of the experiment showing LBR expression level in each cell line relative to wild-type. (E) The above described cell lines were treated exactly as in (C), harvested, and the total cholesterol concentration was determined using a fluorometric assay (see Materials and methods). Values represent a mean of three independent experiments with error bars indicating the standard deviation. A cholesterol standard curve is shown on the right.

in a LBR KO background, indicating that these genetic lesions ultimately result in a failure to sustain cell growth under cholesterol-restrictive conditions.

## The C-terminal sterol reductase domain of LBR is necessary and sufficient for cholesterol production of HeLa cells

We next investigated the functional relationship between the N-terminal nuclear lamin B/chromatin binding domain of LBR and the C-terminal sterol reductase domain (SRD). We generated cell lines in an LBR KO background that expressed either LBR WT, the isolated LBR SRD in an LBR knockout cell background (see *Figure 4A* for LBR domain structure), or the fusion protein Sun2-LBR, which contains the nuclear domain of Sun2 containing a nuclear targeting signal (*Turgay et al., 2010*), fused to the LBR SRD. Additionally, we generated a cell line that expressed an LBR construct that encompasses the nuclear domain of LBR together with the first transmembrane helix of the SRD (LBR TM1), a construct that is competent for INM targeting (*Smith and Blobel, 1993*; *Soullam and Worman, 1993*).

LBR KO-derived cell lines expressing constructs described above were cultured in triplicate in lipoprotein-depleted growth medium for 7 days. Cells were then trypsinized and stained with Trypan blue to exclude non-viable cells and counted. As expected, very few live cells remained in the LBR KO cell sample, while expression of full-length LBR in an LBR knockout cell background restored the cells to near-WT levels of growth (*Figure 5A*). Conversely, LBR TM1 failed to rescue cell growth. Finally, both the isolated LBR SRD and the fusion protein SUN2-LBR rescued the LBR KO phenotype. Given that all constructs were expressed at levels higher than the rescuing WT allele (*Figure 5B*), our results indicate that the SRD is necessary and sufficient for survival of cholesterol-starved cells.

## Disease-associated LBR point mutants substantially reduce NADPH binding

The observations that (i) the LBR N547D and LBR R583Q transgenes are defective in rescuing the cholesterol auxotrophy and that (ii) both mutations map to the SRD, which is necessary and sufficient to complement the LBR KO phenotype, strongly suggests that these mutations affect sterol reductase activity. Indeed, both affected residues are widely conserved in related reductases, including the sterol reductase from *Methylomicrobium alcaliphilum* (maSR1), the structure of which was recently determined (*Li et al., 2015*). Based on a sequence alignment (*Figure 5—figure supplement 1*), the LBR N547 and R583 residues correspond to maSR1 N359 and R395, respectively, both of which map to the NADPH binding pocket, with N359 being implicated in a hydrogen bond formation with a phosphate oxygen of NADPH (*Li et al., 2015*) (*Figure 5C*). We therefore tested whether these mutants displayed a reduced affinity for NADPH, the cofactor that contributes the electrons for the reduction reaction. WT LBR, LBR N547D, and LBR 583Q were purified and their NADPH binding affinities measured using a spectroscopic approach (see materials and methods). WT LBR displayed hyperbolic saturation for the cofactor with a $K_D$ of 24.2 μM. Both mutants had a severely decreased binding affinity to NADPH, with N547D having the strongest effect (*Figure 5D*). We attribute this lack of affinity to a possible charge repulsion between the introduced Asp side chain and the negatively charged phosphate of the cofactor. Given that the physiological concentration of NADPH is in the range of ∼ 100 μM–a concentration range at which the discrepancy of cofactor occupancy of LBR and LBR N547D/LBR R583Q would be maximal– our measurements provide a direct rationale for the inability of those mutants to efficiently rescue the cholesterol auxotrophy (cf. *Figure 4B*). As expected, neither LBR N547D nor LBR 583Q transgenes efficiently restored *de novo*

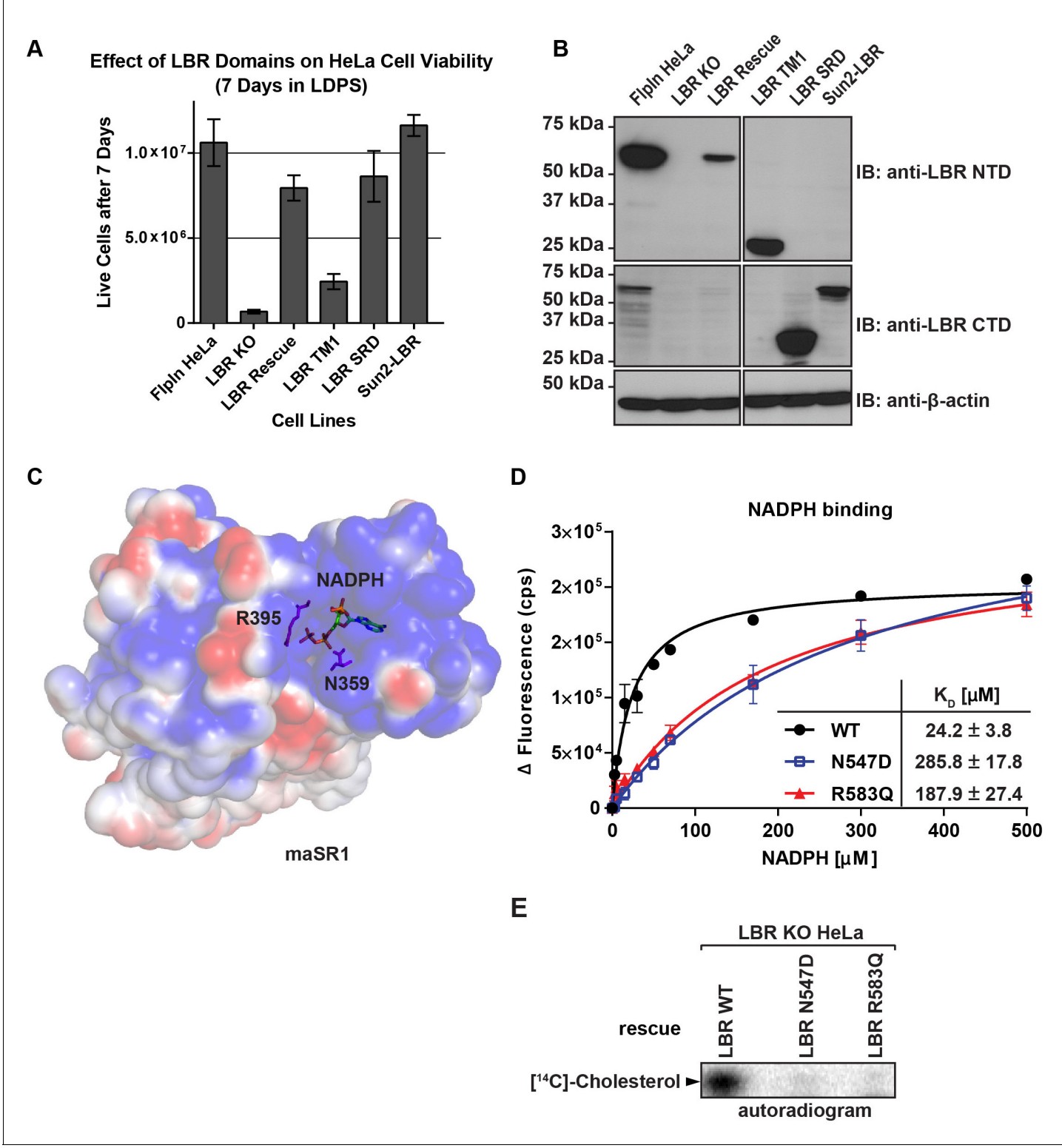

**Figure 5.** The C-terminal sterol reductase domain of LBR Is necessary and sufficient for cell viability under cholesterol restrictive growth conditions. (**A**) WT HeLa cells, LBR knockout HeLa cells, or LBR knockout cells expressing either WT LBR, the LBR nuclear domain plus the first transmembrane helix (LBR-TM1), the LBR sterol reductase domain (LBR-SRD), or the nuclear domain of Sun2 fused to the sterol reductase domain of LBR (Sun2-LBR) were grown in triplicate under cholesterol restrictive growth conditions for 7 days. The cells were then trypsinized and counted and the results were plotted with standard deviations shown. (**B**) Anti-LBR immunoblots of lysates from the above treated cells show LBR expression level relative to WT. Membranes were probed with two distinct anti-LBR antibodies recognizing an N- and C-terminal LBR epitope, respectively. (**C**) Electrostatic surface potential

*Figure 5 continued on next page*

*Figure 5 continued*

representation of themaSR1 (crystal structure (PDB: 4QUV) with kT/e ± 1. NADPH and residues N359 and R395 corresponding to LBR disease-associated residues N547 and R583 (cf. *Figure 5—figure supplement 1*) are shown as sticks. (D) Disease-associated LBR point mutants LBR N547D and LBR R583Q show a decreased affinity for NADPH compared to wild-type LBR. Intrinsic tryptophan fluorescence of purified LBR WT and mutants upon NADPH binding was plotted against NADPH concentration and non-linear regressions were fitted in GraphPad Prism. All measurements were performed in triplicate. (E) HeLa LBR KO cells stably expressing LBR WT, LBR mutant N54D or R583Q were cultured in LPDS containing medium for 48 hr prior to metabolically labeling with [$^{14}$C]-acetate. Lipids were extracted and separated by TLC and visualized via autoradiography. Bands corresponding to [$^{14}$C]-cholesterol are marked by an arrowhead.

The following figure supplement is available for figure 5:

**Figure supplement 1.** Sequence alignment of LBR and related sterol reductases.

cholesterol synthesis in LBR KO cells, which was monitored by metabolic incorporation of $^{14}$C acetate into cholesterol (*Figure 5E*).

## C-terminally truncated LBR proteins are rapidly degraded via a proteasome-dependent pathway

Neither of the LBR C-terminal truncation mutant proteins (1402TΔ or 1600*) were detected via immunoblotting using an LBR antibody that recognizes the N-terminal (nuclear) domain of LBR (see *Figure 4D*). Given that these cell lines were engineered to express their WT, LBR N547D, or LBR 583Q transgenes to comparable levels, we speculate that the LBR truncation either leads to non-sense-mediated decay (NMD) of the encoding mRNA (*Kurosaki and Maquat, 2016*), or that the proteins themselves are recognized as aberrant and degraded by a cellular protein quality-control system (*Labbadia and Morimoto, 2015*).

To distinguish between these possibilities, we monitored the metabolic stability of LBR and its mutant derivatives by pulse-chase analysis as described previously (*Rose et al., 2014*). We found that LBR WT, LBR N547D and LBR R583Q are extremely stable (*Figure 6A and B*), indicating that the lack of cofactor binding does not lead to LBR instability. In contrast, both truncated LBR variants were produced at levels comparable to WT LBR at the beginning of the chase period but were rapidly degraded (*Figure 6A and B*). These observations are consistent with a post-translational effect on protein stability and argue against a major contribution of NMD.

To investigate the degradation mechanisms of LBR 1402TΔ and LBR 1600*, we repeated the pulse-chase experiments on a shorter time scale and in the absence or presence of MG132, a potent cell-permeable proteasome inhibitor (*Rock et al., 1994*). We found that both LBR 1402TΔ and LBR 1600* proteins were degraded extremely rapidly, with little or no protein remaining after 30 min (*Figure 6C and D*). Both mutants were significantly stabilized by MG132, with minimal degradation taking place in 60 min after synthesis. We conclude that the degradation of truncated LBR variants depends on the proteasome. Notably, the kinetics of LBR 1402TΔ and LBR 1600* degradation were remarkably rapid, with half-lives of 10–15 min, especially considering that these mutant derivatives are polytopic membrane proteins. As a standard of comparison, the half-life of CFTRΔ506, a mutant variant of the polytopic chloride channel responsible for cystic fibrosis (*Turnbull et al., 2007*) which is widely used as model substrate to study protein turnover, is four hours (*Heda et al., 2001*). Thus, LBR disease variants have significant potential as novel model substrates to study protein turnover.

## Truncated LBR variants are ubiquitylated

Since protein ubiquitylation is a key step in proteasome-mediated protein turnover of misfolded membrane proteins (*Claessen et al., 2012*; *Raasi and Wolf, 2007*; *Vembar and Brodsky, 2008*), we next determined if the rapidly degraded truncated LBR proteins LBR 1402TΔ and LBR 1600* are ubiquitylated. We expressed either FLAG-LBR WT, FLAG-LBR 1402TΔ, or FLAG-LBR 1600* together with HA-Ubiquitin (Ub) in a LBR knockout cell background. Cells were then treated with either MG132 or vehicle for 2 hr and subjected to denaturing detergent lysis to disrupt non-covalent protein-protein interactions. Following dilution with SDS-free buffer, extracts were immunoprecipitated with an anti-FLAG resin to retrieve tagged LBR protein.

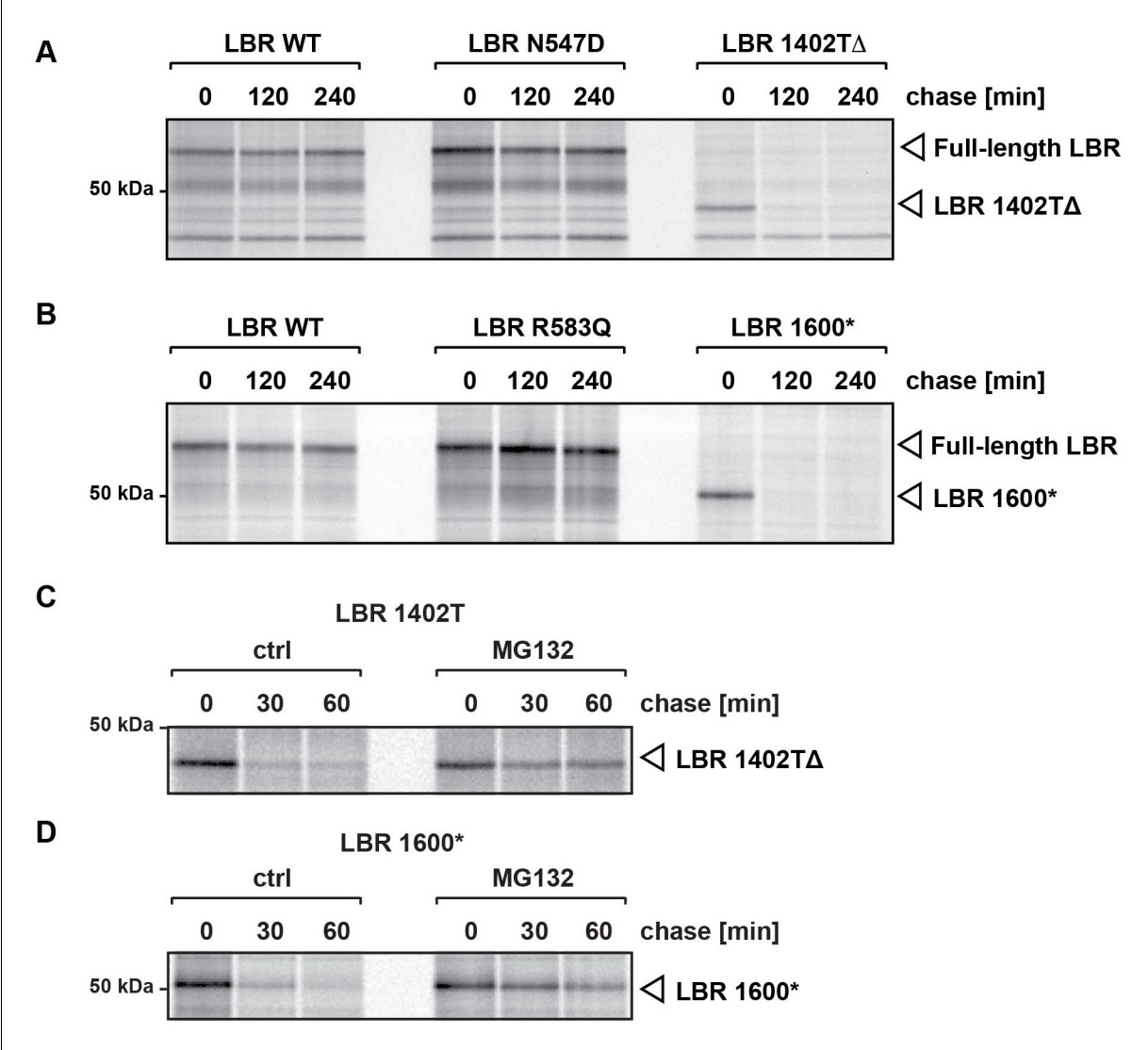

**Figure 6.** C-terminally truncated LBR mutants associated with Pelger-Huët anomaly and Greenberg skeletal dysplasia are rapidly degraded via the proteasome. (A), (B) LBR KO cells expressing either WT LBR or the disease-associate LBR mutants were metabolically labeled with $^{35}$S and then chased with an excess of unlabeled cysteine/methionine. LBR was then retrieved at the indicated time points via immunoprecipitation, resolved by SDS-PAGE and imaged via autoradiography. (C), (D) Turnover of LBR 1402TΔ and LBR 1600* was measured on a shorter time scale in the absence or the presence of MG132.

Anti-HA immunoblotting of input samples demonstrated that HA-Ub was expressed in all HA-Ub transfected cells, with the expected increase of HA-Ub conjugates in the higher molecular mass range in response to proteasomal inhibition (*Figure 7A*, top panel). Identical samples were subjected to immunoblotting with anti-FLAG antibodies to detect LBR or its mutant derivatives. In agreement with our pulse-chase data, both LBR 1402TΔ and LBR 1600* are expressed at extremely low levels under steady-state conditions when compared to LBR WT (*Figure 7A*, bottom panel).

Next, the anti-FLAG immunoprecipitates were subjected to SDS-PAGE and immunoblotting. As expected, HA-Ub conjugated species were significantly more abundant for both LBR 1402TΔ and LBR 1600* than for LBR WT in both untreated and MG132-treated samples, with the latter condition leading to an additional increase (*Figure 7B*, upper panel). Since the levels of unmodified LBR WT far exceed those of LBR 1402TΔ and LBR 1600* (*Figure 7B*, lower panel), these data reflect a disproportionate increase in abundance of ubiquitylated LBR species for the disease-associated variants

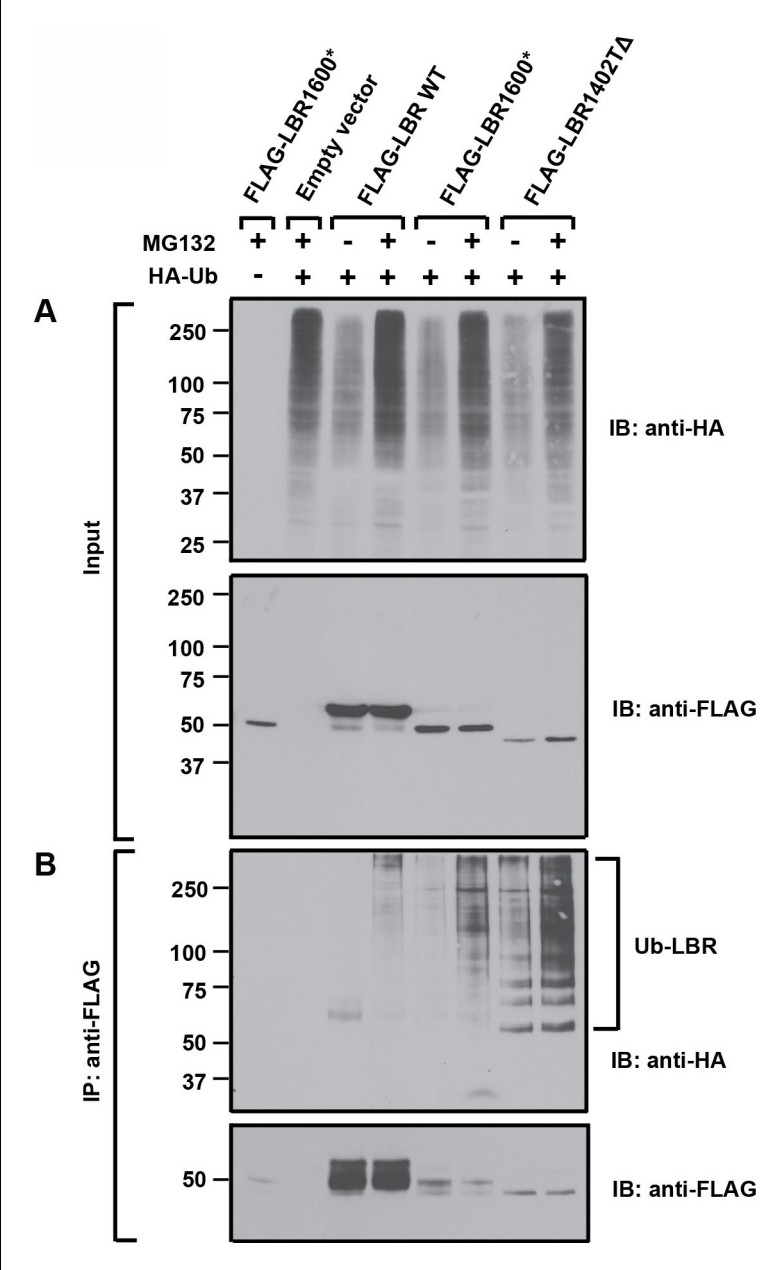

**Figure 7.** LBR 1402TΔ and LBR 1600* proteins are polyubiquitylated. LBR KO cells were co-transfected with plasmids encoding HA-tagged ubiquitin and FLAG-tagged LBR WT, LBR 1600*, or LBR1402TΔ. Sixteen-hours-post transfection, cells were treated with MG132 or DMSO for another 4 hr before harvesting and denaturing SDS lysis. (**A**) Five percent of cell lysates used per immunoprecipitation (**B**) were separated by SDS-PAGE and subjected to immunoblotting using the indicated antibodies (**B**) Lysates were diluted in SDS-free buffer and subjected to immunoprecipitation using anti-FLAG antibody, followed by SDS-PAGE and immunoblotting analysis using the indicated antibodies.

relative to LBR WT. Together, the data argue that the metabolic instability of LBR 1402TΔ and LBR 1600* is caused by degradation via the Ub/proteasome system (UPS).

## Truncated LBR accumulates inside the nucleus after proteasome inhibition

Given our finding that C-terminally truncated LBR is degraded via a UPS-dependent pathway, LBR 1402T and LBR 1600* could represent typical substrates for the pathway responsible for the degradation of ER-resident proteins referred to as ER-associated degradation (ERAD) (*Claessen et al., 2012*; *Raasi and Wolf, 2007*; *Vembar and Brodsky, 2008*). The ERAD machinery can act on LBR 1402TΔ and LBR 1600* before they arrive at the INM. Alternatively, a distinct, INM resident pathway might mediate LBR 1402TΔ and LBR 1600* turnover.

As a first step towards resolving this question, we monitored the cellular localization of LBR 1402TΔ and LBR 1600* in the absence or presence of MG132 using confocal fluorescence microscopy. In the absence of MG132, LBR 1402TΔ is partitioned between the ER and the NE, as judged by a co-staining with anti-LBR and anti-Lamin A/C antibodies (*Figure 8A*, left panel). LBR 1600*, on the other hand, displays nuclear rim staining that is indistinguishable from LBR WT (*Figure 8B*, left panel; cf. *Figure 2A*, upper left panel) (*Worman et al., 1988*).

Against all expectations, both truncated LBR proteins appear to accumulate in the nucleoplasm upon proteasomal inhibition (*Figure 8A and B*, right panels). In addition, both LBR mutants are clearly excluded from nucleoli in the presence of MG132. Quantification of these results, based on a total projection average of 20 independent z-stack series for each condition, is shown in *Figure 8C* (see *Figure 8—figure supplement 1* for a corresponding z-stack series). We observe that with both mutants, MG132 treatment resulted in a shift from a non-nuclear to a nuclear LBR immunofluorescence signal, with LBR 1402TΔ showing a slightly larger shift in localization than LBR 1600*, which resides primarily in the nuclear compartment even in the absence of MG132 (*Figure 8A,B*).

These unexpected results indicate that disease-associated, C-terminally truncated LBR mutants surprisingly accumulate in the nuclear compartment, rather than the ER or cytosol, after MG132 treatment. Given that LBR1600* is extremely short-lived but mainly localizes to the INM under steady-state conditions even in the absence of MG132, we suggest that it is here that membrane dislocation occurs.

The requirement of the AAA+ ATPase p97 for the extraction or dislocation of membrane proteins from the ER membrane is firmly established (*Ye et al., 2003*). Given that p97 has additional functions in the nuclear compartment (*Dantuma and Hoppe, 2012*), we asked whether p97 is implicated in the extraction of LBR 1600* from the INM. LBR KO cells were transfected either with FLAG-LBR 1600* alone or in combination with either p97 WT or p97 QQ, a dominant-negative mutant of p97 that potently blocks p97-dependent functions (*Ernst et al., 2009*; *Ye et al., 2003*). We then performed pulse-chase analyses to monitor the stability of LBR 1600*. As expected, p97 WT had no effect on LBR 1600* degradation (*Figure 8D and E*). However, p97 QQ inhibited LBR 1600* turnover to an extent that is comparable to proteasomal inhibition (*Figure 8D and E*).

We next asked whether p97 QQ can block LBR 1600* turnover in the nuclear compartment using confocal microscopy. Cells were again co-transfected with LBR 1600* and either p97 QQ or p97 WT and then treated with MG132 or carrier. This experimental setup allowed us to explore the possibility of an epistatic relationship between p97 QQ and MG132. If p97 is indeed required for the dislocation of LBR 1600* from the INM, we would expect p97 QQ to prevent the accumulation of LBR 1600* in the nucleoplasm upon MG132 treatment. Using double-staining with anti-p97 and anti-LBR antibodies, we indeed found this to be the case. In the presence of both p97 WT and MG132, we observed the expected LBR 1600* accumulation in the nucleoplasm and to a lesser degree, the ER (*Figure 8F*), whereas LBR 1600* accumulates nearly quantitatively at the nuclear rim in the presence of p97 QQ even in the presence of MG132. The data are consistent with a blockage of the membrane dislocation step at the INM (*Figure 8F*).

In conclusion, these results are consistent with a mechanism in which only a small percentage of LBR 1600* is degraded by the canonical ERAD pathway. The majority of the truncated protein is localized to the INM, where it appears that a second quality control pathway operates to extract and destroy LBR 1600* in a series of reactions employing both p97 and the UPS (*Figure 9*).

## Discussion

In this study, we assign human LBR to an essential role in cholesterol synthesis. We found LBR to be essential for survival under cholesterol-restrictive growth conditions in three different human cell types

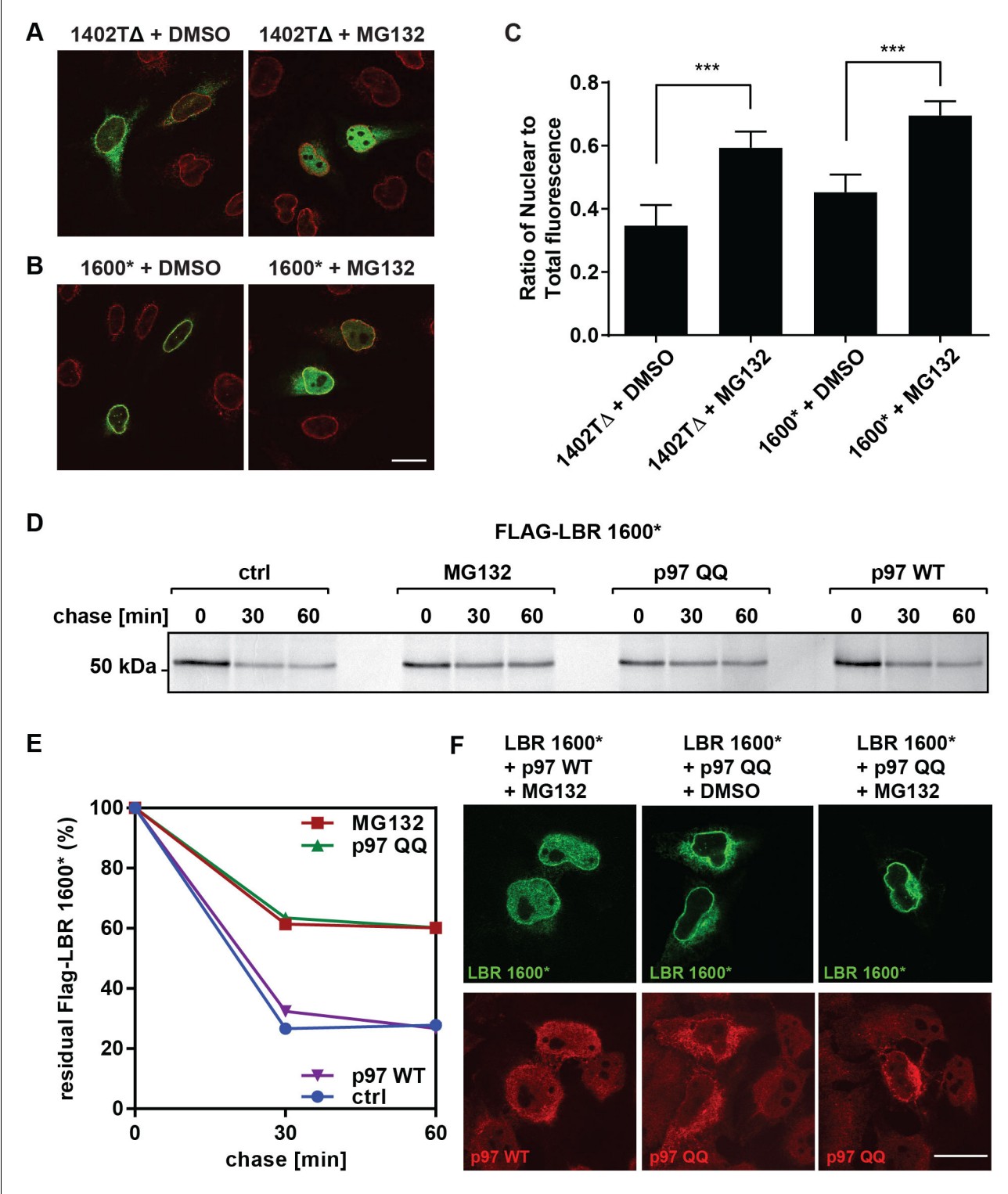

**Figure 8.** LBR 1402TΔ and LBR 1600* proteins accumulate in the nucleus after MG132 treatment. (**A,B**) LBR knockout cells expressing either LBR 1402TΔ or LBR 1600* were treated with MG132 or DMSO vehicle for 4 hr and then fixed, stained, and imaged by confocal fluorescence microscopy. The cells were stained with antibodies against LBR (green) and Lamin A/C (red). (**C**) Cells were treated and imaged as above (**A,B**), and the nuclear fluorescence obtained from 20 complete z-stack series for each condition was quantified using ImageJ, summed up, and standardized relative to the sum of total fluorescence. The ratio of nuclear to total cellular fluorescence is given as arithmetic mean value. Error bar represents mean ± SD. The statistical significance is determined by unpaired t-test. ***p<0.001 (**D**) Pulse-chase analysis of LBR KO HeLa cells co-transfected with FLAG-tagged LBR 1600*

*Figure 8 continued on next page*

*Figure 8 continued*
mutant and p97 WT or QQ mutant. (E) Densitometric quantification of pulse-chase data. (F) LBR-knockout HeLa cells were co-transfected with LBR mutant 1600* and with p97 WT or QQ mutant and treated with 10 μM of MG132 or DMSO for 4 hr. Cells are stained with anti-LBR (green) and anti-p97 (red), and imaged with a confocal microscope. Scale bar = 20 μm.
The following figure supplement is available for figure 8:

**Figure supplement 1.** Complete confocal z-stack series corresponding to *Figure 8A, B*.

(*Figures 2C*, *3A–D*). Correspondingly, LBR-deficient HeLa cells cannot efficiently synthesize cholesterol (*Figure 2D*), but are readily rescued by the addition of LDL particles or free cholesterol to the cholesterol starvation medium (*Figure 2C*). These results are unexpected since previous studies in mice reported that the sterol reductase activities of LBR and TM7SF2 are functionally redundant, suggesting that HEM dysplasia is likely a laminopathy that is unrelated to the sterol reductase activity of LBR (*Wassif et al., 2007*). In cultured HeLa cells, however, LBR does not play a significant role in NE organization despite being a constitutively expressed, abundant INM component (*Figure 2A,B*), possibly due to functional redundancy with some of the multitude of other lamina-associated membrane proteins (*Hetzer and Wente, 2009*; *Schirmer and Gerace, 2005*). In our opinion, a major role for TM7SF2 in generic cholesterol synthesis is difficult to reconcile with the absence of an overt phenotype upon deletion of TM7SF2 in mice, which are not compromised in cholesterol synthesis (*Bennati et al., 2008*). In fact, TM7SF2 is expressed under cholesterol-restrictive conditions in all tested human cell lines (*Figure 3I*), but cannot compensate for the absence of LBR. Nevertheless, it is striking that mice lacking a fully functional LBR allele due to homozygous mutations at the ichthyosis locus (*ic/ic*) are viable, although these animals display numerous phenotypic abnormalities including alopecia, syndactyly and hydrocephalus as well as an increase in embryonic lethality (*Shultz et al., 2003*). Growth defects were also observed in primary cells isolated from mouse models with mutations in the *LBR* gene (*Subramanian et al., 2012*; *Verhagen et al., 2012*). Of note, defects in neutrophil maturation in *ic/ic* animals can be recued in vitro by expression of the LBR SRD (*Subramanian et al., 2012*).

It is likely that LBR function is subject to diversification in the course of evolution. For example, human LBR can rescue sterol reductase deficiency in yeast (*Silve et al., 1998*), while LBR from *Droshophila melanogaster* does not complement this phenotype (*Wagner et al., 2004*). Since *D. melanogaster* is a cholesterol-auxotrophic organism, we speculate that additional LBR functions could involve the N-terminal Tudor domain. While we found this domain to be dispensable for cholesterol synthesis (*Figure 5A*), a possible role could involve heterochromatin organiziation (*Solovei et al., 2013*), which may be related to the nuclear abnormalities observed in e.g. Pelger-Huet anomaly (*Hoffmann et al., 2002*) and mouse models of ichtyosis (*Shultz et al., 2003*). Since the LBR SRD is neccesary and sufficient to restore cholesterol synthesis (*Figure 5A*) and rescues a defect in neutrophil maturation observed in *ic/ic* cells in vitro (*Subramanian et al., 2012*), a knock-in of either the SRD or the Tudor domain (including the first transmembrane domain) into the LBR locus could help to deconvolute distinct functions of the SRD and Tudor domains in mammalian development.

Given that our study establishes LBR as the major sterol reductase required for cholesterol synthesis in human cells, the question arises why a second, functionally equivalent enzyme is encoded in the human genome. Since cholesterol synthesis can proceed through differential, tissue-specific usage of distinct yet interchangeable segments of the established synthesis pathways (*Mitsche et al., 2015*), we speculate that TM7SF2 could be required for cholesterol synthesis in a specific physiological context. Interestingly, a recent study suggested that TM7SF2 may participate in healing of burn wounds (*Lei et al., 2016*), possibly representing a case of a more specialized physiological role.

Based on our finding that all tested disease-causing mutations in LBR fail to complement the cholesterol auxotrophy imposed by an LBR deficiency in our tissue culture model (*Figure 4*), we believe that animal models of LBR malfunction warrant additional scrutiny from the perspective of cholesterol metabolism. The administration of a low cholesterol diet or inhibitors of cholesterol synthesis in animal models with *LBR* mutations could reveal previously unknown phenotypes and thus provide new insight into disease etiology. Similarly, it would be interesting to explore states in which specific cell types are exposed to conditions of low environmental cholesterol in a physiological context, as

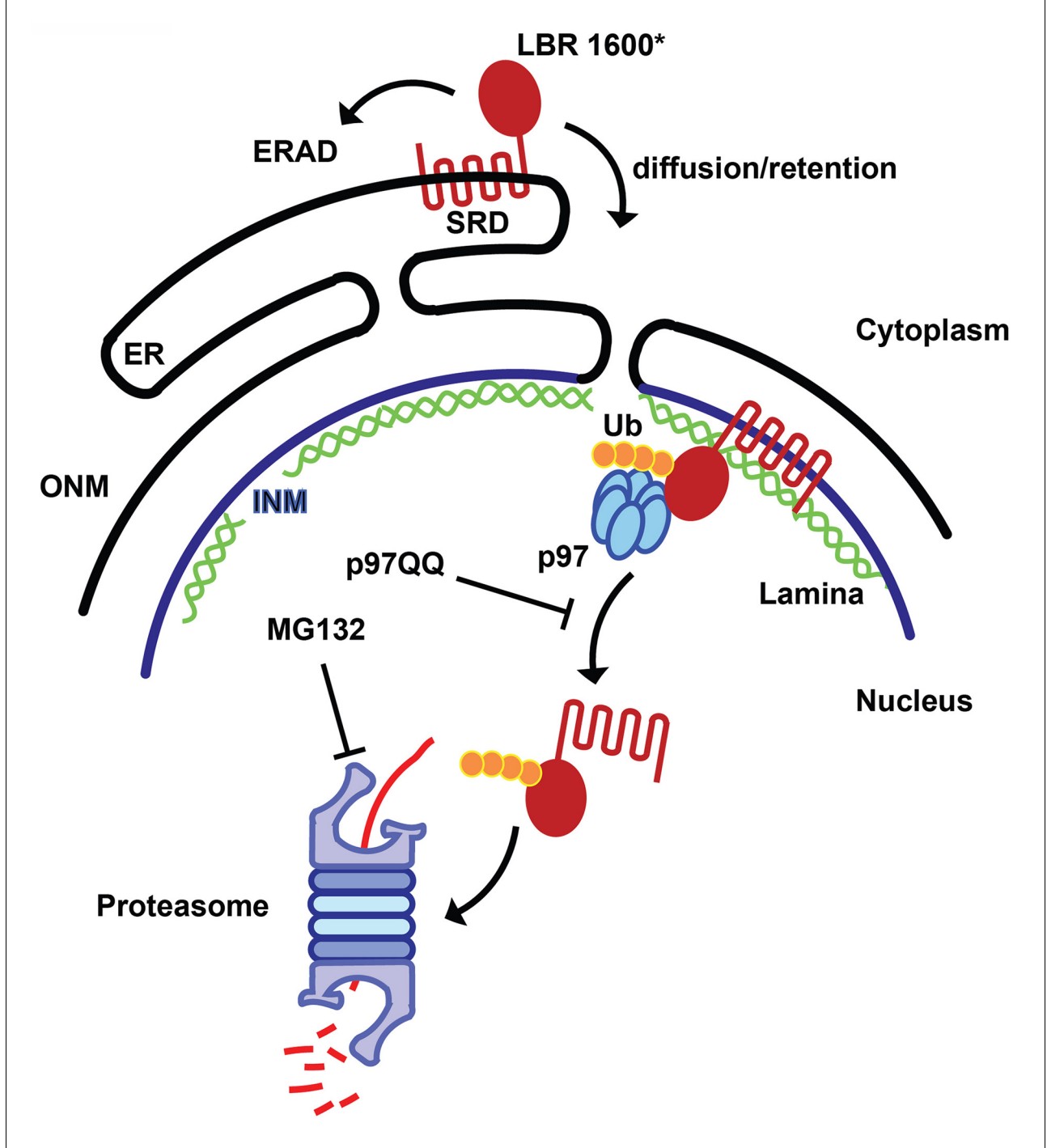

**Figure 9.** Model for partitioning of metabolically unstable LBR variants between ER- and INM-resident protein turnover pathways. C-terminal truncation of LBR (e.g. in LBR 1600* or 1400TΔ) causes the sterol reductase domain (SRD) to misfold, leading to LBR ubiquitylation, membrane dislocation and its subsequent degradation via the Ub/proteasome pathway. A minor portion of LBR1600* is degraded by the canonical ERAD pathway in the ER, whereas the majority will enter the nucleus by virtue of the correctly folded N-terminal domain and be retained by binding to the nuclear lamina. Here, a presumably ERAD-independent pathway is operative, involving both p97 and the Ub/proteasome. Turnover at the INM can be inhibited at the dislocation or degradation step by a dominant negative p97 variant (p97 QQ) or the proteasome inhibitor MG132, respectively. Note that several ERAD components as well as p97 cofactors are omitted for clarity. ER, endoplasmic reticulum; INM, inner nuclear membrane; ONM, outer nuclear membrane; Ub, ubiquitin.

for example in the course of embryogenesis, which could help to rationalize the embryonic-lethal phenotypes in Greenberg skeletal dysplasia.

Another unexpected outcome of this study is that C-terminally truncated LBR proteins found in both Pelger-Huët anomaly and Greenberg skeletal dysplasia are rapidly degraded by a proteasome-dependent protein quality-control pathway that appears to be distinct from the canonical ERAD pathway (*Figures 8*, *9*). Given that INM-resident proteins are synthesized in the ER prior to their targeting to the INM (*Lusk et al., 2007*; *Ungricht and Kutay, 2015*), we propose that the correctly folded N-terminal moiety of LBR containing INM targeting information (*Smith and Blobel, 1993*; *Soullam and Worman, 1993*) leads to the rapid trafficking of LBR 1600* to the INM before elements of the ERAD pathway can act (*Figure 8*). Considering that near-quantitative nuclear accumulation of an HA-tagged LBR variant is achieved in ~20 min (*Ungricht et al., 2015*), and that the half life of LBR 1600* is ~10–15 min, it follows that a significant fraction of LBR 1600* escapes the ERAD system leading to localization to the INM mediated by the intact N-terminal domain, which confers lamin binding (see. *Figure 8B* and *Figure 8—figure supplement 1* demonstrating INM localization of LBR 1600* under steady-state conditions). Upon arrival of LBR 1600* at the INM, a system analogous to but distinct from the known ERAD system is responsible for the ubiquitylation of LBR 1600* and its p97-dependent extraction from the INM (*Figures 8*, *9*). Based on the unusually rapid kinetics and substantial protein accumulation in a dislocated nuclear state in presence of MG132, we believe that the INM degradative system is different from the established ERAD machinery.

In fact, it is still unclear which Ub ligases account for protein turnover at the INM of mammalian cells. In general, the mechanisms of protein quality control operative at the NE of budding yeast are far better understood than analogous systems in mammalian cells (*Deng and Hochstrasser, 2006*; *Foresti et al., 2014*; *Khmelinskii et al., 2014*; *Rose and Schlieker, 2012*; *Webster et al., 2014*). The absence of suitable model substrates in higher eukaryotes has been the major limitation.

This study establishes the long sought-after methodological framework to investigate NE-directed quality control mechanisms in human cells. We propose that the combination of (i) the extremely short half-life of LBR1600*, (ii) its near-exclusive localization to the INM as well as (iii) and ease with which LBR1600* can be arrested at distinct stages of dislocation and turnover make this LBR variant an ideal tool to pursue mechanistic studies aimed at the elucidation of protein quality control and turnover at the INM.

# Materials and methods

## Cell lines

All cell lines and primary cells were purchased from ATCC (HFF-1: ATCC cat# SCRC-1041; HEK293T: ATCC cat# CRL-11268; HeLa: ATCC act# CCL-2) and regularly tested to be Mycoplasma-negative as judged by the absence of extranuclear DAPI staining.

## Generation of LBR knockout HeLa cell lines

LBR knockout HeLa cells were generated using the CRISPR/Cas9 genome editing system as described previously (*Mali et al., 2013*; *Turner et al., 2015*). The CRISPR guide sequence LBR 5'-GACTCCCTCGGCGTCTGGAAGGG-3' targeting the first exon of LBR was chosen from a published index of human exon gRNA targets (*Mali et al., 2013*). Potential knockout colonies were harvested, expanded, and screened both by genotyping PCR and by immunoblotting. The genotyping primers used were LBR gt-F: 5'-TTCAAGCTCTGTTCC-3' and LBR gt-R: 5'-TGTGTATGTATTGACTC-3'; GAPDH-F: 5'-CGACCGGAGTCAACGGATTTGGTCG-3' and GAPDH-R: 5'-GGCAACAATATCCAC TTTACCAGA-3'.

## Illumina MiSeq sequencing

Illumina MiSeq of LBR knockout cell CRISPR target sites was performed in collaboration with the Yale Center for Genomic Analysis. Genomic DNA from LBR knockout FlpIn HeLa cells was harvested using QuickExtract DNA Extraction Solution (Epicentre, Madison, WI) according to manufacturer instructions. A 500 nucleotide region centered around the CRISPR target site of the *LBR* gene (see *Figure 2—figure supplements 1* and *2*) was then PCR amplified using the primers LBR FW 5'-TAG TGTCACATAGATAACGCAGTGGCT-3' and LBR RV 5'-CAAGAGCTCAATCCTCTGCCTTCA-3'. The

resulting mixture was then gel purified and submitted for a single lane of Illumina MiSeq sequencing, obtaining several million reads of the target region. Reads with complete sequence coverage of the target area were binned according to the mutation detected, resulting in the delineation of three separate LBR gene edits, corresponding to the three copies of the *LBR* gene found in HeLa cells.

## Electron microscopy

Electron microscopy was carried out by the Yale Biological Electron Microscopy facility as described previously (*Rose et al., 2014*).

## Generation of LBR WT and LBR mutant rescue cell lines

Stable cell lines expressing the gene of interest under doxycycline control were generated using the FlpIn T-REx cell system (Invitrogen) as described previously based on HeLa cell obtained from ATCC CCL-2 (*Turner et al., 2015*). LBR N547D, LBR R583Q, and LBR 1402TΔ constructs were generated via quickchange mutagenesis according to standard protocols. LBR 1600*, LBR ND, LBR TM1, and LBR SRD were generated by PCR amplification of the relevant cDNA region of LBR as follows: LBR 1600* amino acids 1–534, LBR ND amino acids 1–209, LBR TM1 amino acids 1–246, and LBR SRD amino acids 197–616. The Sun2-LBR fusion construct was generated via fusion PCR and encompasses amino acids 1–177 of Sun2 fused to amino acids 197–616 of LBR.

## Cholesterol synthesis

HeLa cells were grown on a 6-well plate and starved for 48 hrs in DMEM medium containing lipoprotein-deficient serum (LPDS). Cell were metabolically labeled with 2 uCi/well [$^{14}$C]-acetate (Perkin Elmer) for 4 hr at 37°C as described previously (*Zelcer et al., 2014*). Cells were lysed and saponified. The lipids were extracted three times with 2 ml hexane and dried under nitrogen stream. Extracts were re-dissolved in 60 μL hexane and aliquots were separated on a Silica Gel 60 F254 plate (Merck) with a mobile phase of hexane: diethyl ether: glacial acetic acid (60:40:1, v/v/v) as described by *Gill et al. (2011)*. The TLC plate was exposed to an imaging plate (Fujifilm) and visualized with a Storm Scanner (GE Healthcare).

## Real-time PCR

Wild-type or LBR KO HeLa cells were seeded in a 12-well plate at a density of $1\times10^5$ cells/well and transfected with 50 nM of control siRNA or SMARTpool siRNA targeting to SREBP2. After 48h, the cells were split 1:2 into two 12-well plates in normal medium or LPDS medium and incubated for another 2 days. Total RNAs were then isolated and transcribed into cDNA using SuperScript II reverse transcriptase (ThermoFisher Scientific). The RT reactions were diluted 1:5 with water, and 1.25 μL were used in real-time PCR which is carried out using iQ SYBR Green mix and CFX Real-Time PCR Detection System (Bio-Rad). Data was analyzed using △△Ct method, in which the △Ct was calculated first as Ct of internal control (RPL32) was subtracted from each sample, and the △△Ct was further calculated by subtracting △Ct of control group from △Ct of each treated group, and final results were represented as $2^{(-\triangle\triangle Ct)}$. Primer sequences used in qPCR are listed as follows (5' to 3'): RPL32 (Forward: CGGCGTGCAACAAATCTTACTGTGCCG; Reverse: CCAGTTGGGCAGC TCTTTCC), SREBP2 (Forward: CCGGGCGCAACGCAAAC; Reverse: CGCCCATGACACCCGACAA), LBR (Forward: AGTATAGCCTTCGTCCAAGAAGA; Reverse: CAAAGGTTCTCACTGCCAGTT), TM7SF2 (Forward: AACTCAGGCAATCCGATTTACG; Reverse: GGGTCGCAGTTCACAGAAATA), HMGCR (Forward: AGGGGATGCCATGGGGATGA; Reverse: ACGGCTAGAATCTGCATTTCAGGG)

## Cloning of TM7SF2 cDNA

The TM7SF2 cDNA was cloned from HeLa cells by RT-PCR. Total RNAs were isolated from HeLa cells cultured in cholesterol-restrictive medium and reversed transcribed into cDNA using a cDNA amplification kit (SMARTer RACE kit, Clontech). The TM7SF2 cDNA was amplified using a universal forward primer (5'-CTAATACGACTCACTATAGGGC-3') and a gene-specific reverse primer (5'-TCAGTAGA TGTAGGGCATGATGCG-3'). The resulting PCR product was subjected to Sanger sequencing.

## Antibodies and immunoblotting

Immunoblotting was performed according to standard protocols in 5% (wt/vol) skim milk in Tris-buffered saline and 0.1% (vol/vol) Tween 20 (TBS-T) using Western Lightning plus ECL reagent (Perkin Elmer). The antibodies used in this study were the following (numbering according to http://antibodyregistry.org): anti-LBR N-terminal domain (AB_775968, Abcam) at 1:2,000, anti-LBR C--terminal domain (AB_10712378, Abcam) at 1:2,000, anti-Tubulin (AB_477583, Sigma) at 1:2000, anti- β-actin (AB_306371, Abcam) at 1:2000, and anti-TM7SF2 (Covance custom antiserum). The specificity of the anti-TM7SF2 antibody was confirmed by inclusion of a TM7SF2 knockout cell lysate as a reference (see *Figure 3I*).

## Immunofluorescence microscopy

Immunofluorescence microscopy was performed as described previously (*Rose et al., 2014*). The primary antibodies used were the following (numbering according to http://antibodyregistry.org): anti-LBR N-terminal domain (AB_775968, Abcam) at 1:500, anti-Lamin B1 (AB_10107828, Abcam) at 1:500, anti-Lamin A+C (AB_306913, Abcam) at 1:500, anti-β-actin (AB_306371, Abcam) at 1:1,000, anti-Sun1 (AB_1080462, Sigma) at 1:500, anti-Sun2 (Covance custom antiserum) at 1:1,000, anti-Lap1 (Covance custom antiserum) (*Turner et al., 2015*)(*Zhao et al., 2013*) at 1:1,000, anti-Mab414 (AB_448181, Abcam) at 1:500, anti-hnRNP A1 (AB_305145, Abcam) at 1:500, anti-hnRNP A2B1 (AB_732978, Abcam) at 1:500, and anti-calnexin (AB_1310022, Abcam) at 1:500.

## Cholesterol-starvation cell growth assay

200,000 HeLa cells were counted in triplicate, washed with 1mL PBS, and then resuspended in lipoprotein-depleted growth medium (DMEM + 10% lipoprotein depleted fetal bovine serum) in 24-well plates. The cells were then cultured using standard mammalian tissue culture conditions for 7 days. All samples were split 1:3 on days 2 and 4. If exogenous cholesterol or LDL was used in the experiment, it was introduced on day 2 and was continued through the end of the experiment. If expression of LBR from the FlpIn locus was used (e.g. for LBR rescue), all cell samples were treated with doxycycline (500 ng/mL) beginning on day 1 and continuing through the end of the experiment. After day 4, the cells were grown until day 7, at which time they were trypsinized, treated with 1:1 trypan blue to exclude nonviable cells, and then counted in triplicate.

Human foreskin fibroblast cell (HFF) and 293T were seeded in a 12-well plate at a density of $1\times10^5$ cells/well one day prior to transfection. Cells were transfected with 50 nM of control siRNA or ON-TARGETplus SMARTpool targeting to LBR (GE Dharmacon) using Lipofectamine RNAiMax (ThermoFisher Scientific). After 48h, cells were trypsinized, washed with PBS, and split 1:3 (293T) or 1:5 (HFF) into 24-well plates in medium containing LPDS, and incubated for another 1 day (293T) or 3 days (HFF), respectively. The surviving adherent cells were quantified using crystal violet staining as described (*Zivadinovic et al., 2005*).

## Total cholesterol detection assay

Total cholesterol measurements of cell extracts were performed using a fluorometric cholesterol + cholesterol ester detection kit (Abcam ab65359). Triplicate sets of cells treated for 7 days as described above and samples were subjected to fluorometric total cholesterol detection according to manufacturer instructions.

## NADPH binding assay

Flag-tagged LBR and mutants were expressed in Expi293 cells (Life technologies) for 3 days as described by the manufacturer and purified as reported previously (*Zhao et al., 2013*). The NADPH binding measurement was performed using intrinsic tryptophan fluorescence of LBR. In short, 150 µl of 10 µM purified LBR was titrated by increasing concentration of NADPH in elution buffer (20 mM HEPES, 150 mM NaCl, 5 mM $MgCl_2$, 5 mM KCl, 0.01% (w/v) n-dodecyl-D-maltoside (DDM) pH 7.5). Fluorescence decrease (Ex:295 nm / Em:335 nm) upon NADPH addition was recorded on a spectrofluorometer (Photon Technology International) at 20°C and further corrected for the inner filter effect (*Woodyer et al., 2005*). The fluorescence change was plotted against NADPH concentration and non-linearly fitted using GraphPad Prism.

## Acknowledgements

We thank Morven Graham for assistance with electron microscopy and Jonathan Rodenfels, members of the Yale Nucleus Club, and the Schlieker laboratory for comments on the manuscript. This work was supported by NIH (1DP2OD008624-01).

## Additional information

### Funding

| Funder | Grant reference number | Author |
|---|---|---|
| NIH Office of the Director | 1DP2OD008624-01 | Christian Dirk Schlieker |

The funders had no role in study design, data collection and interpretation, or the decision to submit the work for publication.

### Author contributions

P-LT, CZ, ET, CS, Designed research, performed experiments, Analyzed data, Wrote the manuscript, Contributed unpublished essential data or reagents

### Author ORCIDs

Christian Schlieker, http://orcid.org/0000-0002-1237-4267

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
