## [Decision Letter]

Thank you for submitting your article "The Lamin B Receptor is Essential for Cholesterol Synthesis and Perturbed by Disease-causing Mutations" for consideration by *eLife*. Your article has been favorably evaluated by Vivek Malhotra (Senior editor) and three reviewers, one of whom is a member of our Board of Reviewing Editors.

The reviewers have discussed the reviews with one another and the Reviewing Editor has drafted this decision to help you prepare a revised submission

All three of the reviewers liked this manuscript and thought that, with suitable revisions, it could be appropriate for publication in *eLife*. However, all three reviewers were insistent that revisions, including additional experiments, are necessary. One reviewer commented that the quality of the writing in this paper is excellent, but the same reviewer was disappointed by the content of the Discussion section.

Summary:

The authors have shown that the lamin B receptor is not required for maintenance of nuclear structure, but that it is required for cholesterol synthesis-at least in HeLa cells. They carefully characterize lamin B receptor mutants associated with human disease. Finally, they clearly show proteosomal turnover of the lamin B receptor in the nucleus. Their findings help to elucidate mechanisms for turnover of inner nuclear membrane proteins.

Essential revisions:

1) It is important to determine whether the importance of the lamin B receptor is unique to HeLa cells. Do HeLa cells express TM7SF2 transcripts and protein?

A survey of LBR and TM7SF2 transcript and protein levels in several different human cell lines would be instructive. Are there coding mutations in TM7SF2 in HeLa cells? *eLife* readers need to have some inkling about whether the essential nature of the LBR for cholesterol synthesis is a peculiarity of one tumor cell line.

2) Document an accumulation of δ 14 sterols in the LBR-deficient cells.

3) Measure cholesterol synthesis in LBR-deficient cells.

4) It would be highly desirable to show that the mutants are ubiquitinylated at more native levels of expression; protein overexpression could induce proteasomal degradation.

5) It would be desirable to take a fresh look at SREBP control of LBR and TMSSF2 expression.

6) While the quality of the writing was excellent, the Discussion section was too short and disappointing in terms of the content. Not to discuss the ichthyosis phenotype in the mice and speculate about its etiology was odd. Not to include a discussion of the Wassif paper that concluded that the enzymatic activities of the two proteins were redundant was odd. Not to mention conclusions from the mouse KO of the DHCR14 gene is odd. Not to discuss in more detail the Subramanian paper was odd. Putting your data in the context of earlier contributions is the right thing to do.

---

## [Author Response]

*1) It is important to determine whether the importance of the lamin B receptor is unique to HeLa cells.*

We generalized our findings in several cell types as suggested by this reviewer. We chose HEK 293T cells given that they are commonly used in many laboratories as well as human foreskin fibroblasts (HFFs) to include non-transformed primary cells in our analysis. Both LBR-silenced HEK 293T and HFFs exhibit significant cell death relative to non-targeting siRNA controls under cholesterol-restrictive conditions, corroborating our assignment of LBR to an essential function in cholesterol synthesis. These data are now included as an additional figure (new Figure 3).

*Do HeLa cells express TM7SF2 transcripts and protein?*

*A survey of LBR and TM7SF2 transcript and protein levels in several different human cell lines would be instructive.*

This was indeed a deficiency in our original manuscript. We therefore raised antibodies against TM7SF2, allowing us to demonstrate that TM7SF2 is expressed at very low levels under normal growth conditions. TM7SF2 expression is induced when cells are cultured in lipid-depleted growth medium, which we now demonstrate on the transcript and protein level via qPCR and immunoblotting analysis, respectively (new Figure 3, Figure 3—figure supplement 1). As outlined below (see point #5), we have also added data demonstrating that the transcriptional upregulation depends on SREBP-2 (Figure 3—figure supplement 1). Interestingly, different cell types/lines display differences in their TM7SF2 expression, consistent with the recent demonstration of tissue- and cell type-specific differences in cholesterol synthesis (Mitsche et al., e*Life*. 2015 Jun 26;4:e07999. doi: 10.7554/*eLife*.07999).

Are there coding mutations in TM7SF2 in HeLa cells? eLife readers need to have some inkling about whether the essential nature of the LBR for cholesterol synthesis is a peculiarity of one tumor cell line.

We share the concern of this reviewer. Two major TM7SF2 transcripts are annotated in NCBI (http://www.ncbi.nlm.nih.gov/gene/7108), specifying two distinct TM7SF2 isoforms of 46 and 43 kDa, respectively Using a cloning strategy based on RT-PCR in conjunction with an unbiased (not relying on sequence homology to isolate/amplify the 5'-end) 5' RACE approach, we were able to isolate/amplify the cDNA for the longer isoform 1. This isoform does not have a single mutation. Given that the TM7SF2 protein in HeLa cells co-migrates (on SDS gels) with TM7SF2 proteins detected in HFFs and other cell lines (new Figure 3), we conclude that the longer isoform 1 is the major TM7SF2 gene product in several cell types. The sequencing results are included as new [Supplementary-material SD1-data]. Please note also the newly added data showing that HFFs and HEK 293s similarly rely on LBR to sustain cholesterol synthesis (new Figure 3). Thus, the essential role of LBR in HeLa cells cannot be attributed to their degeneracy or to a coding mutation in TM7SF2.

*2) Document an accumulation of δ 14 sterols in the LBR-deficient cells.*

We feel that repeating previously published data is not needed to validate our conclusions for the following reasons: (i) the enzymatic activity of LBR is firmly established via genetic complementation and biochemical experiments, (ii) δ 14 sterols have been detected in several cell types in LBR-deficient ichthyosis mouse models (e.g. J Immunol 188(71):85-102.). (iii) Most importantly, δ 14 sterols were detected in cultured cells of a HEM/Greenberg skeletal dysplasia patient (Am J Hum Genet. 72(3):1013-7). Of note, this fetus carried the same mutation that we characterize in our study. This is the most physiologically relevant demonstration of δ 14 sterol accumulation resulting from LBR mutations that we can think of. We have now included these above-cited references in our revised Discussion, and hope that the reviewers/editor agree that the implementation of the highly specialized GC/MS-based assay required to measure δ 14 sterols is not necessary to validate our conclusions, even more so given that we have now added direct proof for a lack of de novo cholesterol synthesis in LBR-deficient/perturbed cells to the revised manuscript (new Figure 2 and Figure 5).

In fact, we feel that our demonstration of the degradation of mutant variants and our demonstration and structural rationalization of a cofactor binding defect allows us to arrive at a more precise definition of the molecular defects associated with the disease-causing mutations. These data are in perfect agreement with the reported accumulation of δ 14 sterols upon LBR perturbation in patient tissues/cells and mouse tissues; therefore, we consider these studies and our work as being perfectly complementary to each other (we note that none of these studies looked at metabolic instability of LBR mutants, cofactor affinity, or provided a structural rationale for disease mutations). We therefore hope that the reviewers/editors agree that our revised study significantly advances our molecular understanding of disease etiology.

*3) Measure cholesterol synthesis in LBR-deficient cells.*

We reported in the original version of the manuscript that total (or steady-state) cholesterol levels are reduced by ~30% in LBR knockout or mutant cell lines after the starvation period. We believe that this range reflects a viability threshold, i.e. cell death ensues at lower concentrations of cholesterol, making it impossible to accurately quantify this effect and to deconvolute the contribution of de novo synthesis. We have therefore implemented metabolic labeling techniques as suggested by the reviewer to directly measure de novo synthesis using ^14^C acetate in conjunction with thin layer chromatography/autoradiography. We are most grateful for the reviewer's suggestion since the resulting data are a clear “black and white” result and are even more convincing than the original data. LBR-deficient cells or those that express disease mutants are in fact completely deficient in de novo synthesis of cholesterol (new Figure 2 and Figure 5). Given our newly added data demonstrating that HeLa cells express TM7SF2 devoid of coding mutations in an LBR deficient background, these data reinforce our central conclusion that LBR is essential for cholesterol synthesis despite the presence of TM7SF2.

*4) It would be highly desirable to show that the mutants are ubiquitinylated at more native levels of expression; protein overexpression could induce proteasomal degradation.*

We share the reviewer’s general concern about overexpression artifacts. Indeed, most of the data in this study are based on carefully controlled expression levels using defined genetic loci for expression of transgenes at near-endogenous expression levels. However, we want to point out that much of the published literature in the protein turnover/ERAD field relies on strongly overexpressed proteins since it is otherwise difficult to achieve robustly detectable levels of metabolically unstable proteins. Please note that neither LBR1600* nor LBR1400T are detectable under steady-state conditions in immunoblots (cf. Figure 4) since they are rapidly degraded, forcing us to increase expression by using a strong promoter to reach the detection limit of immunoblots in our ubiquitin-relevant experiments. Please also note that even when overexpressed, the levels of LBR1600* and LBR1400T are significantly lower than endogenous LBR (see Figure 10 in which we compare endogenous levels of LBR to the input lysates used for the IP experiments in Figure 7). Given that overexpressed LBR WT, which we included as control, is not degraded at much higher levels of expression (Figure 6) and displays only negligible levels of ubiquitination (Figure 7), we hope the reviewer will agree that overexpression cannot account for the ubiquitination or the metabolic instability of LBR mutant derivatives. We have also repeated the pulse-chase experiments that were originally based on transient transfection of LBR1600* and LBR1400T into LBR KO cells. Our new data, which are based on stable cell lines expressing of LBR1600* and LBR1400T at levels below the detection limits of immunoblots (cf. Figure 4), faithfully recapitulate our previous result (i.e. rapid turnover/MG132-sensitivity). We have therefore replaced the original panels in Figure 6/D by these new panels representing our recent data obtained at near-endogenous/lower expression levels.

Author response image 1.**DOI:**
http://dx.doi.org/10.7554/eLife.16011.020

*5) It would be desirable to take a fresh look at SREBP control of LBR and TMSSF2 expression.*

We followed the reviewer's suggestion and tested whether LBR and TM7SF2 expression is dependent on SREBP2 in HeLa cells. In accordance with the published literature (Biochim Biophys Acta. 2006 Jul;1761(6):677-85), we found that TM7SF2 expression is induced in response to cholesterol starvation (cf. point #1), while LBR is constitutively expressed and not significantly up-regulated (new Figure 3, Figure 3—figure supplement 1). As expected, induction of TM7SF2 expression depends on SREBP2, as judged by a significant reduction of TM7SF2 expression in SREBP2-silenced cells relative to a non-targeting siRNA control. Similar results were obtained for HMGCoAR, LDM and NSDHL. We have not included all of these data in the manuscript since the revision has increased the size of manuscript tremendously, and a detailed analysis of transcriptional control is outside the scope of our study. We also generalized our findings by including HFFs and HEK 293T cells in this analysis (new Figure 3).

*6) While the quality of the writing was excellent, the Discussion section was too short and disappointing in terms of the content. Not to discuss the ichthyosis phenotype in the mice and speculate about its etiology was odd. Not to include a discussion of the Wassif paper that concluded that the enzymatic activities of the two proteins were redundant was odd. Not to mention conclusions from the mouse KO of the DHCR14 gene is odd. Not to discuss in more detail the Subramanian paper was odd. Putting your data in the context of earlier contributions is the right thing to do.*

We agree with the reviewer's assessment and have expanded the Discussion section of manuscript to rectify this deficiency. In the revised version, we place our findings in the context of the literature that was omitted in our original version. We also included a number of additional recently published studies that are relevant for the interpretation of our data.